# Weighted Multi-Prompt Learning with Description-free Large Language Model Distillation

**Sua Lee**[1]*, **Kyubum Shin**[2]*, **Jung Ho Park**[1]
[1]Seoul National University,    [2]Naver AI

## Abstract

Recent advances in pre-trained Vision Language Models (VLM) have shown promising potential for effectively adapting to downstream tasks through *prompt learning*, without the need for additional annotated paired datasets. To supplement the text information in VLM trained on correlations with vision data, new approaches leveraging Large Language Models (LLM) in prompts have been proposed, enhancing robustness to unseen and diverse data. Existing methods typically extract text-based responses (i.e., *descriptions*) from LLM to incorporate into prompts; however, this approach suffers from high variability and low reliability. In this work, we propose **De**scription-free **Mul**ti-prompt Learning(**DeMul**), a novel method that eliminates the process of extracting descriptions and instead directly distills knowledge from LLM into prompts. By adopting a description-free approach, prompts can encapsulate richer semantics while still being represented as continuous vectors for optimization, thereby eliminating the need for discrete pre-defined templates. Additionally, in a multi-prompt setting, we empirically demonstrate the potential of prompt weighting in reflecting the importance of different prompts during training. Experimental results show that our approach achieves superior performance across 11 recognition datasets.

## 1 Introduction

What are the sentences that "best" describe the *Golden Retriever* in the image shown in Fig. 1? Even when familiar with the dog, answers to this question will always differ, as there cannot be definitive correct answers. Thus, if such ambiguous sentences are defined as the "categories" of the image, it becomes challenging for others to categorize it accurately.

Recently, there has been a growing interest in leveraging pre-trained Large Language Models (LLM), such as GPT(Brown et al., 2020), across various tasks. In image recognition, the *descriptions* that LLM responded to the query, e.g. {`"What are useful features for distinguishing a class name?"`}, have been used to enhance accuracy compared to using only a single class label. Notably, in vision language models (VLM), these descriptions are adeptly integrated into prompts, which are then utilized as categories for classification. However, even with well-pretrained LLM, there are limitations, including **high variability** in responses and **low reliability** of some descriptions. These limitations occur because: (i) the query is inherently open-ended, leading to multiple plausible interpretations, (ii) the format of the query influences the responses, and (iii) inherent biases affect the generated descriptions.

Formally, aside from applying descriptions, *prompt learning* has been studied as an efficient method to enhance generalization in pre-trained VLM, such as CLIP(Radford et al., 2021), GLIP(Li et al., 2022), and ALIGN(Jia et al., 2021), without the necessity of additional task-specific annotated data. To set the text prompt used in these models, the standard approach involves simply applying a pre-defined template, e.g., `"A photo of a {class}."`. However, trivial variations in this template, such as `"a"` or `"."`, can have a profound impact on inference performance(Zhou et al., 2022b). To mitigate this variation, the prompts can be defined as learnable continuous vectors that can be optimized, so that each prompt can be trained to have optimal arrangements and semantics.

---

*Equal contribution. Correspondence to: `sualee.susan@gmail.com`

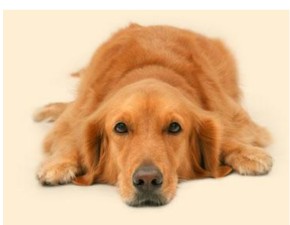

- 0: Four-legged domestic dog breed
- 1: Medium to large-sized dog breed
- 2: Golden or cream-colored fur
- 3: Floppy ears
- 4: Muscular build
- 5: Friendly expression
- 6: Often seen with a tennis ball or frisbee7:
- 7: May have darker fur on face or ears

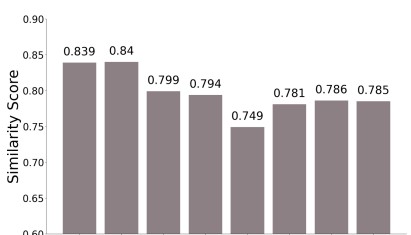

Figure 1: **High variability and low reliability of GPT-based descriptions:** The example of descriptions obtained by asking GPT the question, `"What are useful features for distinguishing a Golden Retriever?"`. Some descriptions highlight highly distinctive visual features, while others convey ambiguous meanings and often begin with qualifiers such as 'often', 'may', or 'maybe' that can reduce clarity. Moreover, it is uncertain whether the last description accurately portrays the characteristics of a Labrado Retriever. Comparing the text similarity between these descriptions and the class name "Labrado Retriever" reveals significant variability in reliability. There is also a noticeable discrepancy between our manual assessment of the descriptions and the determination of useful features for classification based on their similarity.

With prompts no longer fixed to a single template, multiple prompts can be assigned to each class, empirically demonstrating superiority over single prompts. This raises the question: *Which prompt holds relatively significant semantics?* While there has been made effort to handle the distribution of existing learnable prompts, the method dealing with the importance of prompts has yet to be studied.

In this work, we present **Description-free Multi-prompt Learning (DeMul)** as a way to directly distillate the LLM's pre-trained distribution without descriptions. Specifically, instead of directly inserting descriptions into prompts, our approach map learnable prompts into the LLM embedding space and distill them to absorb the semantics.

We chose GPT-based embedding models as the LLM to distill, which are accessible through the APIs provided by OpenAI. The public API available for transferring GPT is divided into two main types(Balkus & Yan, 2022): the *Completion Endpoint* which is a text-based conversational model, and the *Embedding Endpoint* which uses embeddings with more reliable performance. While existing description-based methods are limited to using the Completion Endpoint, we first employ the Embedding Endpoint, enabling description-free distillation. By leveraging this API, we can handle prompts as embedding vectors instead of text, eliminating the need for pre-defined templates and allowing them to be optimized as CoOp(Zhou et al., 2022b) explored. Additionally, since prompts are learnable vectors, the importance of the semantics they contain continuously changes during training. DeMul introduces prompt weighting to reflect this variation in importance.

To summarize our contributions:

1. We propose a **description-free distillation** approach that removes the process of extracting descriptions and instead directly distills the pre-trained knowledge of LLM. The learnable prompts are mapped into the LLM embedding space, where they are optimized to capture meaningful semantics.

2. In a multi-prompt setting, the semantics that each prompt learns and their corresponding importance dynamically change during training. We introduce **prompt weighting** to adjust this importance, and our experiments demonstrate that this approach benefits the learning process.

3. We utilize CLIP, one of the most extensively researched models for prompt learning in VLM, as our baseline. We compared its performance across a total of 11 datasets. Ours demonstrates superior results in most datasets, surpassing the existing baselines using description-based methods or learnable prompt optimization methods.

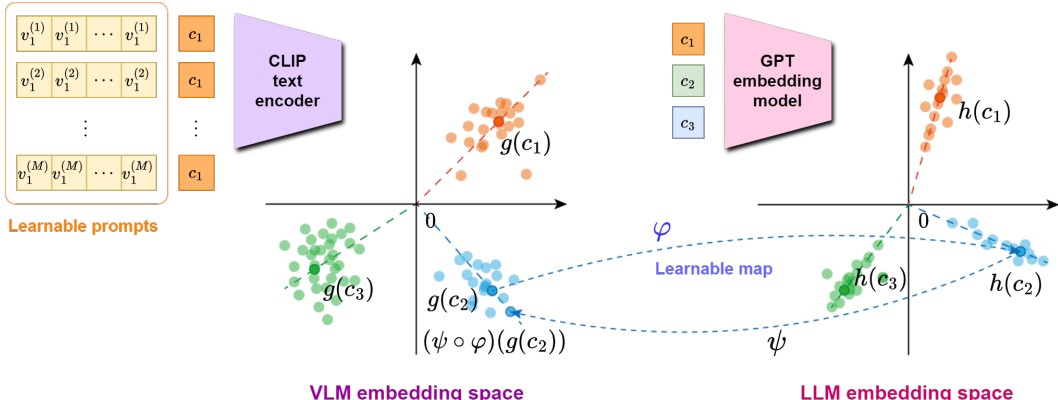

Figure 2: **An overall framework of DeMul:** Here, $c_*$ denotes each class, $g$ represents the CLIP text encoder, and $h$ represents the GPT embedding model. The learning objective is to develop learnable prompts and a function $\varphi$ that captures the semantics of GPTs. Initially, the learnable prompts are randomly generated but are then updated and trained to minimize the angular difference with the GPT embedding vectors of each class. Trained with a mapping loss, $\varphi$ initially preserves the embedding orientation of the set of classes, but is subsequently trained to maintain the directions of the class embeddings adjusted to the prompts.

## 2 PRELIMINARIES AND BACKGROUND

In this section, we provide a concise overview of zero-shot Vision-Language Models (VLM) using hand-crafted prompts, specifically focusing on CLIP. We then describe CoOp, a few-shot method for prompt learning that automatically optimizes prompt templates. Our method applies to various models and tasks that use text embeddings.

**Contrastive Language-Image Pretraining (CLIP)**  CLIP is trained on a web-scale dataset comprising image-text pairs to learn aligned representations through contrastive learning. Specifically, it incorporates two modality-specific encoders—an image encoder $f(\cdot)$ and a text encoder $g(\cdot)$—which are trained to ensure that the correct pairs of embeddings are closely matched in the joint embedding space. Consider an input $(x, y)$, where $x \in \mathbb{R}^{C \times H \times W}$ and $y \in \mathbb{R}^K$. The normalized feature vectors are denoted as $z = f(x)/\|f(x)\|_2$ for the image and $w_y = g(p_0(y))/\|g(p_0(y))\|_2$ for the text $y$, where $p_0(y)$ is the prompt generated for the class $y$ using a pre-defined template, `"A photo of a {class}."`. The prediction probability of $x_i$ is calculated as shown in Eq. 1, where $\tau$ is the temperature parameter of the softmax function.

$$\mathbb{P}(y|x) = \frac{\exp(z^\top w_y / \tau)}{\sum_{i=1}^{K} \exp(z^\top w_i / \tau)} \tag{1}$$

**Context Optimization (CoOp)**  CoOp is a pioneering method to efficiently adapt the context of prompts for downstream visual recognition tasks. Instead of using pre-defined discrete tokens, CoOp employs $N$ learnable continuous parameters $\{v_1, v_2, \cdots, v_N\}$ as the template to be optimized, where each $v_i$ vector has the same dimension as the word embeddings. The learnable vectors $V = \{v_i\}_{i=1}^N$ are shared across all classes, and the generated prompts for each class $c$ are defined as $t = p_*(V, c) = [v_1][v_2] \cdots [v_N][c]$, where prompting $p_*$ is a concatenation function of consecutive vectors and the class name. The learnable tokens are optimized using few-shot samples by maximizing the matching score between the text and image embeddings.

## 3 METHOD

In this section, we present DeMul and its key components. We propose a method for extracting information from an LLM into learnable prompts without relying on hand-crafted descriptions.

Additionally, we introduce a weighted multi-prompt learning approach using importance sampling to perform few-shot recognition with a fixed number of prompts effectively.

## 3.1 DISTILLING LARGE LANGUAGE MODEL WITHOUT MANUAL DESCRIPTIONS

The key question in our proposed approach is how to distill directly the GPT text semantics without hand-crafted descriptions. Specifically, *Can we infuse class-specific information learned by GPT into a prompt without explicit descriptions?* To achieve this, instead of querying GPT to extract features related to a class, we aim to train prompts to have maximized semantic correlations with class names within the GPT embedding space. (The overall workflow for better understanding is illustrated in Figure 2.) GPT models were originally developed for text generation tasks, but they are also utilized for text similarity-related tasks (e.g., text search, code search, sentence similarity, text classification). These models take text inputs and output 3072-dimensional vectors, having learned from large-scale language data to effectively measure similarities between various texts. While the exact training data, architecture, and other details are not publicly disclosed, this approach has demonstrated superior performance across many LLM-based embedding models. In our study, we employed the `text-embedding-3-large` model, which achieved a 64.6% accuracy in MTEB(Muennighoff et al., 2022) eval.

**Mapping prompts**   To leverage the semantic potential of the GPT embedding space for visual prompts, we developed a mechanism for aligning the CLIP embedding vectors with the GPT space. Since 5-layered MLPs are nonlinear and are not diffeomorphism in general, we focus on preserving the direction of the embedding vectors. The function $\varphi$ maps from the CLIP embedding space into the GPT embedding space and $\psi$ maps from the GPT embedding space back to the CLIP embedding space such that $\psi \circ \varphi$ forms a conformal map on a set of points. Throughout this paper, we refer to this conformal map as a *cyclic mapping*.

Since the few-shot recognition task requires a small amount of training data, providing nice initial values can increase the learning stability of the model. Thus, both $\varphi$ and $\psi$ are pre-trained on a comprehensive dataset $\mathcal{D}_{\text{name}}$ that contains common class names as well as class names that appear in the benchmark dataset. $\mathcal{D}_{\text{name}}$ consists of class names from WordNet and 12 other datasets used in the section 4.1, encompassing a wide array of semantic contexts to establish robust initial mappings.

As the fine-tuning process of $\varphi$ progresses with $\psi$ frozen, the set of directions preserved by the cyclic mapping $\psi \circ \varphi$ changes from $\mathcal{D}_{\text{name}}$ to a dataset $\mathcal{D}_{\text{mapping}}$ of learnable prompts. To learn cyclic mapping, we propose the following loss function which measures how closely the cycled embeddings resemble the directions of the original embeddings from the dataset. The similarity loss is formulated as follows:

$$\mathcal{L}_{\text{mapping}} = 1 - \frac{1}{N} \sum_{i=1}^{N} d_{\cos}\left(\psi(\varphi(t_i)), t_i\right) \tag{2}$$

where $t_i$ represents a prompt embedding in $\mathcal{D}_{\text{mapping}}$, $d_{\cos}$ denotes the cosine similarity, and $N$ is the number of data points in $\mathcal{D}_{\text{mapping}}$.

**Distillation in GPT embedding space**   In the multi-prompt setting, for each class $c$, a set of $M$ multiple learnable prompt vectors is defined as $T = \{t_i\}_{i=1}^{M}$, where each $t_i$ is generated by applying the prompting $p_*$ to different learnable vector sets $V_i$ and the class $c$, denoted by $t_i = p_*(V_i, c)$. This approach allows for creating a diverse array of prompts for each class, leveraging multiple vector configurations to capture various semantic nuances of the class. The effectiveness of these prompts is measured using a distillation process in the GPT embedding space. Each prompt $t_i$ is mapped to its GPT embedding by a function $\varphi$, resulting in a set of transformed prompts $\{\varphi(t_i)\}_{i=1}^{M}$. These transformed prompts are then aligned with the corresponding GPT class embeddings $h(c)$, optimized to ensure the correct semantic correlation:

$$\mathcal{L}_{\text{distill}} = -\frac{1}{K} \sum_{i=1}^{K} \left( \frac{1}{M} \sum_{j=1}^{M} \log \mathbb{P}(\varphi(t_{ij}) \mid h(c_i)) \right) \tag{3}$$

where $\mathbb{P}$ is the softmax function of Eq. 1, facilitating the training process by aiming to maximize the probability that each prompt correctly aligns with its respective class in the GPT space.

### 3.2 Weighted Multi-prompt Learning

In the context of visual recognition, the challenge is not only to capture the semantic richness of classes via text prompts but also to ensure effective classification in the CLIP embedding space. In a multi-prompt setting, where each class $c_i$ is associated with multiple prompts $T_i = \{t_{ij}\}_{j=1}^{M}$, the classification loss is initially defined as the average probability overall prompts for a given class:

$$\mathcal{L}_{\text{cls}} = -\frac{1}{K} \sum_{i=1}^{K} \log \left( \frac{1}{M} \sum_{j=1}^{M} \mathbb{P}(y = c_i | x, t_{ij}) \right) \tag{4}$$

This approach, however, does not account for the varying importance of each prompt within a class, as different semantics may contribute differently to the recognition task.

To address this, we introduce a prompt weighting mechanism that dynamically adjusts the importance of each prompt during training, recognizing that the number of relevant semantics or the importance of each semantic can vary significantly between classes. This is particularly crucial because the optimal number of prompts to effectively represent a class is not only non-trivial to determine heuristically but also varies across classes.

Each prompt $t_{ij}$ within the class $c_i$ is assigned a learnable weight $w_{ij}$, reflecting its relative importance. The classification loss for each class is then reformulated to incorporate these weights, providing a weighted average probability that accounts for the differentiated contribution of each prompt:

$$\mathcal{L}_{\text{cls}} = -\frac{1}{K} \sum_{i=1}^{K} \log \left( \frac{1}{M} \sum_{j=1}^{M} w_{ij} \cdot \mathbb{P}(y = c_i | x, t_{ij}) \right) + \lambda \sum_{i=1}^{K} \sum_{j=1}^{M} |w_{ij}| \tag{5}$$

where $\lambda$ is a regularization parameter that controls the trade-off between the classification loss and the L1 penalty. The weights $\{w_{ij}\}$ are normalized for each class, ensuring that $\sum_{j=1}^{M} w_{ij} = 1$. This weighted approach with L1 regularization enhances model flexibility by allowing it to emphasize more informative prompts while diminishing the impact of less relevant ones. The addition of the L1 term encourages sparsity, promoting a scenario where fewer but more significant prompts are actively used, thereby optimizing the classification performance and computational efficiency in processing visual tasks.

### 3.3 Training

The overall training objective for the Weighted Multi-prompt Learning system combines the distillation loss and the weighted classification loss into a total loss function. This total loss is designed to optimize both the semantic alignment in the GPT embedding space and the classification accuracy in the CLIP embedding space. It is formulated as a weighted sum of the two losses:

$$\mathcal{L}_{\text{total}} = \mathcal{L}_{\text{cls}} + \alpha \mathcal{L}_{\text{distill}} \tag{6}$$

where $\alpha$ is a hyperparameter that balances the contribution of the distillation loss and the classification loss. This parameter allows the model to prioritize between the alignment of the prompts with the GPT model's embeddings and the direct classification performance, depending on the specific requirements of the task or the dataset characteristics.

## 4 Experiments

### 4.1 Experiment setup

**Datasets** We evaluate our approach over 11 datasets, including ImageNet(Deng et al., 2009) and publicly available image recognition datasets used in GalLoP(Lafon et al., 2024): SUN397(Xiao et al., 2010), Stanford Cars(Krause et al., 2013), UCF101(Soomro et al., 2012), Caltech101(Li et al., 2017), EuroSAT(Helber et al., 2019), FGVC Aircraft(Maji et al., 2013), Food101(Bossard et al., 2014), DTD(Cimpoi et al., 2014), Oxford Flowers(Nilsback & Zisserman, 2008) and Oxford Pets(Parkhi

---

**Algorithm 1** Training process of DeMul

---

**Require:** Pre-trained VLM encoders of image $f$ and text $g$
**Require:** LLM encoder $h$
**Require:** The number of classes $K$, the number of prompts $M$, the number of epochs $T$, data
  mini-batch size $B$, and prompt mini-batch size $B'$
**Require:** Training dataset $\mathcal{D}_{\text{train}}$ for a target task and the set of class names $\{c_i\}_{i=1}^K$
 1: Compute text embeddings $\{g(c_i)\}_{i=1}^K$ and $\{h(c_i)\}_{i=1}^K$
 2: Randomly initialize the set of prompts $\Theta_p = \{V_j\}_{j=1}^M$
 3: Initialize $\varphi_\theta$ with pre-trained values, $\psi$ frozen
 4: **for** $\tau = 0$ to $T$ **do**
 5:      Sample mini-batches $\left\{(x_b^{(\tau)}, y_b^{(\tau)})\right\}_{b=1}^B$ in $\mathcal{D}_{\text{train}}$ and $\left\{V_j^{(\tau)}\right\}_{j=1}^{B'}$ in $\mathcal{P}$, respectively
 6:      Compute image embeddings $z_b = f(x_b^{(\tau)})$            $(1 \leq b \leq B)$
 7:      Compute prompt embeddings $t_{ij} = g(p_*(V_j^{(\tau)}, g(c_i)))$     $(1 \leq i \leq K$ and $1 \leq j \leq B')$
 8:      Compute LLM embeddings and cyclic embeddings by

$$\varphi_\theta(t_{ij}) \quad \text{and} \quad \psi(\varphi_\theta(t_{ij})) \qquad\qquad (1 \leq i \leq K \text{ and } 1 \leq j \leq B')$$

 9:      Compute the total loss

$$\mathcal{L}_{\text{total}}(\Theta_p) = \mathcal{L}_{\text{cls}}(\Theta_p) + \mathcal{L}_{\text{distill}}(\Theta_p) \qquad\qquad \text{(by the equation (6))}$$

10:     Compute the cyclic mapping loss $\mathcal{L}_{\text{mapping}}(\theta)$           (by the equation (2))
11:     Update parameters $\Theta_p$ and $\theta$ by using $\mathcal{L}_{\text{total}}$ and $\mathcal{L}_{\text{mapping}}$, respectively
12: **end for**

---

et al., 2012). These datasets collectively represent a comprehensive benchmark, covering a diverse set of vision tasks that include classification of general objects, scenes, fine-grained categories, or specialized tasks like recognizing textures, cars and satellite imagery. This extensive variety ensures a robust validation of our approach across a broad spectrum of visual recognition challenges. We follow the same train and test set splits as provided by CoOp(Zhou et al., 2022b) and GalLoP(Lafon et al., 2024), using 1, 2, 4, 8 and 16 shots for training and full test sets for evaluating.

**Baselines** We compare the classification performance of DeMul with the zero-shot or few-shot methods in VLM prompt learning, containing state-of-the-art approach. Our analysis includes zero-shot CLIP(Radford et al., 2021) as a foundational VLM baseline, and extends to two predominant approaches: description-based methods and continuous prompt optimization methods. Among the description-based methods, we compare with multi-prompt dCLIP(Menon & Vondrick, 2022), which incorporates descriptions extracted from GPT into the prompts, and WaffleCLIP(Roth et al., 2023), which further explores prompt template diversity. These approaches typically utilize a zero-shot strategy, focusing on efficient utilization of pre-defined descriptions without further optimization of the prompt template.

In contrast, the continuous prompt optimization methods, exemplified by CoOp(Zhou et al., 2022b), MaPLe(Khattak et al., 2023a), PromptSRC(Khattak et al., 2023b), and GalLoP(Lafon et al., 2024), define prompts as continuous learnable vectors and optimize them within the loss function to enhance the model's performance on classification. This comparative framework allows us to comprehensively assess the effectiveness of DeMul in leveraging both descriptive elements and dynamic prompt optimization to enhance classification accuracy.

**Implementation details** Following existing baselines, we adopt the publicly available pre-trained CLIP model with the visual backbone of ViT-B/16(Alexey, 2020) in all experiments. The number of context tokens $N$ and the number of prompts $M$ are set to 16 and 32, respectively. We optimize the prompts for 100 epochs with SGD optimizer and cosine decay learning rate scheduler, while the base learning rate is set to 0.01 on most datasets. The regularization weight $\lambda$ is set at 0.05, and the loss balance parameter $\alpha$ is established at 0.5. Additionally, to avoid memory overhead, we introduce a prompt sampling strategy used in ProDA(Lu et al., 2022). For each iteration, we randomly selected the batch of prompts During each training iteration, we randomly sample a batch of prompts, and

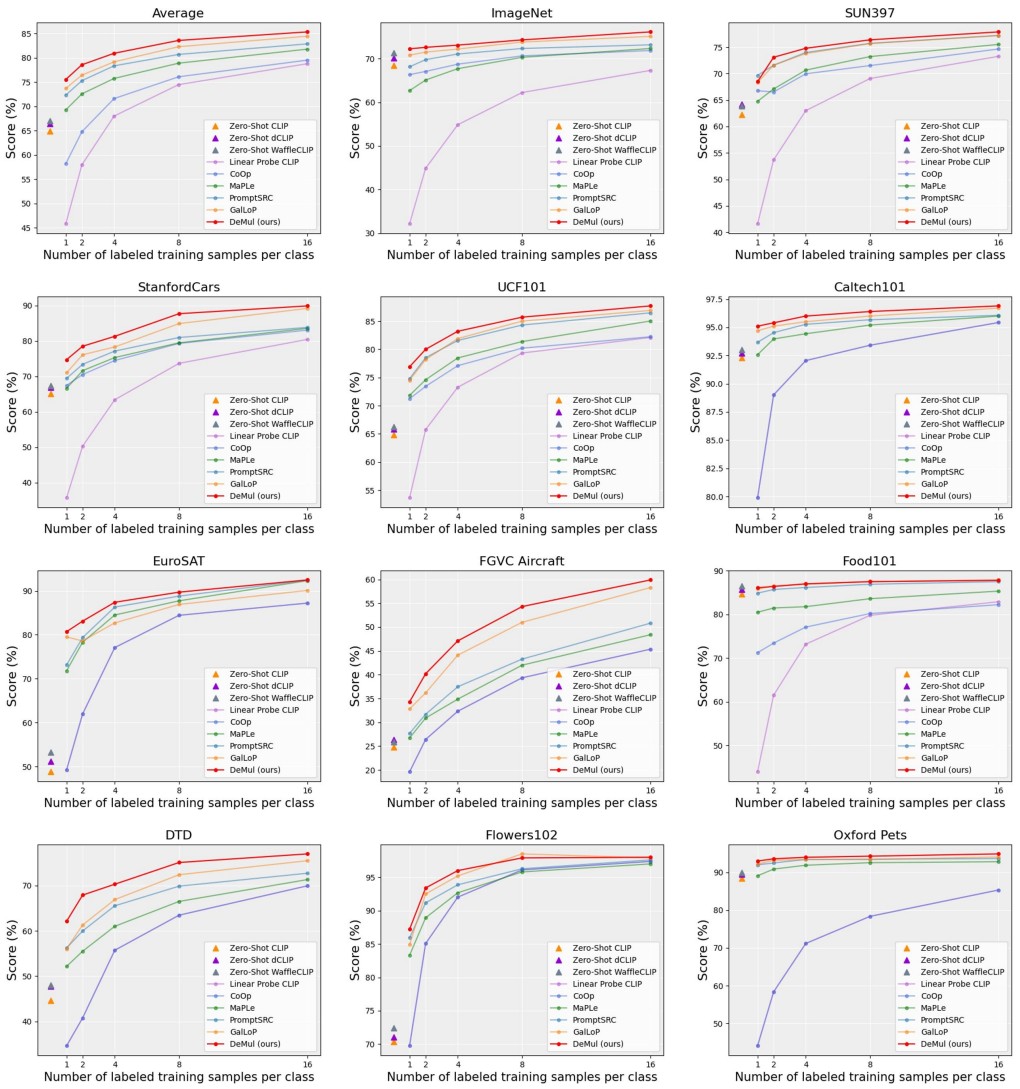

Figure 3: **Comparison of few-shot learning accuracy on the 11 downstream datasets:** Overall, DeMul (indicated by red lines) outperforms CLIP as well as description-based methods (dCLIP(Menon & Vondrick, 2022), WaffleCLIP(Roth et al., 2023)) and continuous prompt optimization methods (CoOp(Zhou et al., 2022b), MaPLe(Khattak et al., 2023a), PromptSRC(Khattak et al., 2023b), GalLoP(Lafon et al., 2024)).

proportionally increase the total number of learning epochs to ensure comprehensive learning. We do experiments on an V100 GPU for datasets with fewer than 100 classes, and utilize an A100 GPU for larger datasets.

## 4.2 PERFORMANCE

Figure 3 summarizes the comparison of DeMul against baselines across 11 datasets and average (detailed scores in Appendix B). A comparative analysis was conducted on few-shot image classification

Top-1 accuracy across downstream datasets, with all pre-trained CLIP models maintained in a frozen state. Our approach demonstrated significant improvements in most settings. On average, compared to the CLIP baseline, our method achieved an 10.5% increase in 1-shot and a 20.3% increase in

Table 1: **Ablation study:** An ablation study on the choice of two distinct losses. (Average Top-1 acc across 11 datasets)

| #(training samples)/class | 1 | 2 | 4 | 8 | 16 |
|---|---|---|---|---|---|
| GalLoP | 73.7 | 76.4 | 79.1 | 82.3 | 84.5 |
| Ours w/o $\mathcal{L}_{\text{distill}}$ | 75.0 | 78.0 | 80.4 | 83.0 | 84.8 |
| Ours w/o weighted | 75.2 | 77.8 | 80.1 | 83.1 | 85.2 |
| Ours | **75.5** | **78.6** | **80.9** | **83.6** | **85.3** |

16-shot accuracy. Even when compared to GalLoP, which exhibits highest performance in learnable multi-prompt optimization, our approach improved by 1.8% in 1-shot and 0.9% in 16-shot. It also shows that DeMul significantly outperforms other zero-shot or few-shot prompt learning methods in most cases. The results suggest that directly training for classification utility, rather than embedding descriptions' semantics into prompts, yields more meaningful benefits.

Additionally, The two distinct loss functions of DeMul($\mathcal{L}_{\text{distill}}$ and prompt weighting) were evaluated separately in the ablation experiment (Table 1). While the performance differences were not substantial, both LLM distillation and prompt weighting consistently demonstrated improvements compared to GalLoP. It also highlights the effectiveness of DeMul when utilizing both methods together.

Table 2: **Study for different LLM models:** DeMul's experimental results in 16-shot when distilling semantic information from different LLM models.

| Method | ImageNet | SUN397 | Stanford Cars | UCF101 | Caltech101 | EuroSAT | FGVC Aircraft | Food101 | DTD | Oxford Flowers | Average |
|---|---|---|---|---|---|---|---|---|---|---|---|
| text-embedding-ada-002 | 74.7 | 76.3 | 88.2 | 86.3 | 95.0 | 90.0 | 57.8 | 86.1 | 75.3 | 95.2 | 82.5 |
| text-embedding-3-small | 75.5 | 77.1 | 89.0 | 86.9 | 95.7 | 91.7 | 58.8 | 87.3 | 76.1 | 96.3 | 83.4 |
| text-embedding-3-large | **76.1** | **77.9** | **89.9** | **87.7** | **96.9** | **92.5** | **59.9** | **87.8** | **77.0** | **98.0** | **84.4** |

Since DeMul is capable of utilizing various pre-trained LLM models, we conducted a few-shot classification experiment using differently trained LLM models. The embedding dimensions for the models `text-embedding-ada-002` and `text-embedding-3-small` are 1536, whereas the embedding dimension for `text-embedding-3-large` is 3072. It has also been reported that both `text-embedding-3-small` and `text-embedding-3-large` are the most recently released models and outperform `text-embedding-ada-002`. The result of the few-shot classification performance in Table 2 reveals a positive correlation between classification performance and the capability of the LLM models.

**Mapping spaces**  In Fig. 4, we analyzed the effectiveness of the mapping function $\varphi$ and its pseudo-inverse $\psi$ during the training process using UMAP for visualization. When examining the initialized 10 prompts in the CLIP space, we can observe that the 10 classes clearly form clusters, as shown in Fig. 4-(a). This clustering occurs because the same set of prompts, labeled with class names, have identical learnable vectors and are therefore positioned closely together. However, if $\psi$ is not frozen and is fine-tuned as suggested in DeMul, the visualization of prompts mapped to the GPT space, as seen in Fig. 4-(b), tends to lose its distinct patterns. Conversely, freezing $\psi$ and fine-tuning $\varphi$ with a consistent loss preserves the clustering tendency in the GPT space, as demonstrated in Fig. 4-(c). Thus, our fine-tuning method allows for the preservation of CLIP space semantics while mapping to GPT space.

**Prompts and weights correlations**  During the training process with five prompts for each class, we examined the changes in prompt weights and the similarity between prompts and class names at five epochs (Fig 5). As the epochs progressed, the prompts increasingly aligned with the class names in terms of similarity, and each prompt exhibited a unique pattern of change. This indicates that each prompt learns different semantics, and the five prompts collectively form clusters, facilitating

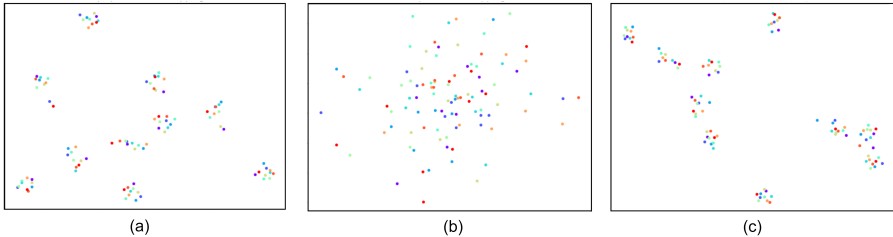

Figure 4: **The U-map visualization of CLIP and GPT embedding spaces** The U-map visualization based on the EuroSAT dataset, which consists of 10 classes, with each class represented by 10 prompts. Prompts belonging to the same class are marked with the same color. **(a)** Initialized prompts in CLIP space, **(b)** Mapped prompts in GPT space, finetuning $\varphi$ without freezing $\psi$, and **(c)** Mapped prompts in GPT space, finetuning $\varphi$ with freezing $\psi$.

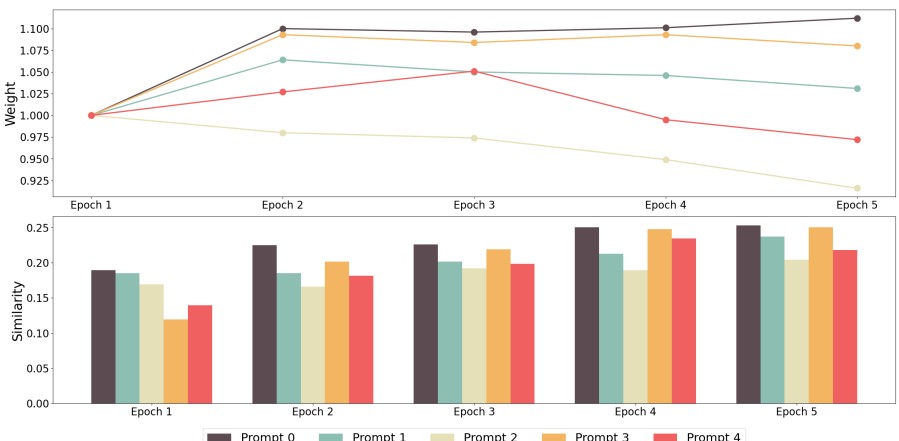

Figure 5: **Variations of the weights of prompts:** The changes were observed at five epochs (0, 15, 30, 45, 60) during the training process. The training process was conducted using the Food101 dataset, which comprises 101 classes, with each class represented by five prompts. **(Top)** illustrates the variations in weights corresponding to each prompt, while **(Bottom)** shows the similarity between each prompt and its respective class names. Both provide average scores across all classes.

effective classification. Additionally, an analysis of the corresponding prompt weights revealed that higher similarities generally correlate with higher weights, and vice versa. Despite the prompts possessing distinct semantics, those closer to the class names significantly influence classification, thus reflecting the importance of prompt weights in the model.

## 5 RELATED WORK

**Vision-language models**   Vision-language models(VLM), which learn rich image-text correlations in the joint embedding space, have shown promising performance in various tasks. VLM trained on web-scaled large datasets can be directly applied to downstream tasks without the necessity of a heavy fine-tuning process. Given image-text pairs data (Schuhmann et al., 2022; 2021), this approach first employs independent encoders that extract meaningful features from text and images, and trains them simultaneously using certain pre-training objectives. Pre-training objectives are mainly categorized (Zhang et al., 2023) into: (1) Generative-objectives (Chen et al., 2022; Li et al., 2021; Ko & Gu, 2022; Yu et al., 2022), which capture meaningful image/text features via learning masked image generating models, (2) Alignment-objectives (Li et al., 2022; Yao et al., 2022; 2021; Zhou et al., 2023), which train to ensure the given text appropriately describe the given image, and (3) Contrastive-objectives (Radford et al., 2021; Jia et al., 2021; Wu et al., 2021; Li et al., 2021; Cui et al., 2022; Yao et al., 2021;

Yang et al., 2022), which are based on contrastive learning methods. The representative model CLIP (Radford et al., 2021) simultaneously trains image and text encoders using contrastive-objectives, by pulling positive samples close and pushing negative samples faraway. The pre-trained CLIP targeting zero-shot prediction has also demonstrated superior performance in application tasks including classic image recognition (Radford et al., 2021; Zhou et al., 2022b), detection (Feng et al., 2022; Maaz et al., 2022; Bangalath et al., 2022), or segementation (Ding et al., 2022; Lüddecke & Ecker, 2022; Rao et al., 2022), etc. Our research aims to enhance recent studies on VLM applications, specifically focusing on facilitating the adaptation and deployment of such models within downstream datasets.

**Prompt learning**  Prompt learning, or prompt engineering(Dosovitskiy et al., 2020; Radford et al., 2019), has emerged recently in NLP as a strategy for effectively transfer large language pre-trained models. It deliberately places mask tokens within sentence templates, then trains the model to predict the word to fill the blank. The model uses the rest of the template to fill it in, highlighting the importance of how the template is designed. Initially, these templates (or prompts) were manually hand-crafted by humans(Brown et al., 2020; Petroni et al., 2019; Radford et al., 2019). Then, to reduce heuristic efforts, advanced approaches automatically search for prompts arranged from discrete tokens(Jiang et al., 2020; Shin et al., 2020), or learn vector-based continuous prompts(Han et al., 2022; Lester et al., 2021; Li & Liang, 2021; Zhong et al., 2021). In CoOp(Zhou et al., 2022b), continuous prompt learning was applied to VLM for the first time. While CLIP uses pre-defined prompts, it has been demonstrated that automatically learning prompts that minimize classification loss can improve performance. As an extension of CoOp, there are methods aimed at enhancing robustness to domain shifts(Zhou et al., 2022a) or learning the distribution of multiple prompts(Lu et al., 2022). Recently, there have been attempts(Menon & Vondrick, 2022; Roth et al., 2023; Pratt et al., 2023; Hu et al., 2023) to utilize Large Language Models(LLM), e.g., GPT-3(Brown et al., 2020), to embed meaningful class-wise context information into prompts. By querying an LLM pre-trained on web-scale datasets, features helpful for distinguishing each class, known as descriptions, can be extracted. These descriptions can be directly inserted into prompts(Menon & Vondrick, 2022), or knowledge distillation can be used to transfer this richer information to continuous vectors in prompt learning(Hu et al., 2023). Building on these concepts, methods like GaLLoP(Lafon et al., 2024), MaPLe(Khattak et al., 2023a), and PromptSRCKhattak et al. (2023b) further explore the potentials of prompt learning by utilizing multiple prompts tailored for specific tasks or layers within a model. GaLLoP, specifically, aims to harness both global and local prompt strategies to optimize the interaction between language and visual elements, significantly enhancing the model's adaptability across diverse scenarios and datasets. MaPLe integrates prompts across several layers of both textual and visual encoders to facilitate deeper integration of language and visual cues. PromptSRC enhances this approach by incorporating several regularization losses which improve not only accuracy but also robustness, allowing for better adaptation to various domain shifts and conditions. These methods underscore the growing sophistication in prompt engineering aimed at maximizing the utility of pre-trained models across increasingly complex scenarios.

# 6 CONCLUSION AND DISCUSSIONS

Our study introduces a novel description-free multi-prompt learning(DeMul) approach, leveraging direct distillation of knowledge from Large Language Models (LLM) into prompts without extracting descriptive text for vision-language models. This method effectively preserves the semantic richness of the prompts while allowing the continuous prompt optimization for few-shot classification tasks. Using pre-trained CLIP, empirical results across 11 downstream datasets confirm that our approach not only simplifies the prompt learning process but also consistently outperforms traditional methods that rely on pre-defined templates or descriptions. This indicates a significant step forward in utilizing pre-trained models for enhancing image classification tasks.

However, there are still some limitations to address. While we have mitigated the high variability of description-based methods, it still depends on the distribution of training images, which can lead to variations in performance due to its few-shot learning framework. Additionally, to maintain memory-efficient training, we have kept the learnable vectors of prompts identical across all classes, as an extension of existing works. This approach can hinder the development of class-specific prompts. For future work, we aim to explore memory-efficient methods that can also robustly handle unseen distributions and still capture class-specific prompts.

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

## A    EXAMPLE OF DESCRIPTIONS

Table 3 shows the examples of class descriptions extracted using the GPT-3 API, previously employed in description-based methods. Consistent with existing works, we employed the `gpt-3.5-turbo-instruct` API for extracting examples. The responses from the API vary significantly with each query, both in content and quantity. Typically, the extracted descriptions accurately depict the visual features of the class, but they often include ambiguous sentences that begin with terms like "may" or "usually", and sometimes list characteristics that are difficult to verify visually (e.g., fried in a pan). Additionally, biases may result in the extraction of descriptions that are irrelevant or incorrect with respect to the class.

Table 3: **The examples of descriptions:** Examples of descriptions for some classes.

**Examples of descriptions**

**Class name: Cheese cake**
1. Creamy white or light brown color
2. Creamy, smooth texture
3. A plate or serving dish in the photo
4. Various toppings such as fruit or chocolate
5. May have a decorative pattern or writing on top

**Class name: French toast**
1. Bread that has been soaked in egg and milk
2. Often served with butter, syrup, or fruit
3. Fried in a pan
4. Sweet and savory taste
5. Typically breakfast food

**Class name: Helicopter**
1. Flying aircraft
2. Metal body with rotors on top
3. Usually painted in bright colors
4. Cockpit for pilot and passengers

**Class name: Chair**
1. Furniture piece
2. Wooden, metal, or plastic material
3. Four legs or a base for support
4. Sometimes has armrests
5. Various designs and styles

**Class name: Rose**
1. Flowering plant
2. Colors of red, pink, or yellow
3. Thorny stems and leaves
4. Sweet, floral scent
5. Grows in clusters on a green stem

**Class name: Bath towel**
1. A piece of fabric
2. Usually white or brightly colored
3. Used for drying oneself after a bath or shower
4. May be hung on a towel rack or hook
5. May be folded or rolled up

**Class name: Baby**
1. Small human being
2. Soft, delicate skin
3. Round head with large eye, small nose
4. Small, chubby limbs
5. Often wearing diapers or clothing
6. Can make facial expressions and vocalizations

**Class name: Truck**
1. Large, heavy motor vehicle
2. Typically has four wheels
3. Usually has a long, rectangular bed for carrying cargo
4. Can come in a variety of colors and designs
5. May have a logo or company name displayed on the side

# B   MAIN RESULTS DETAIL

Table 4 presents the detailed results of DeMul and various baseline methods across 11 datasets. All experiments were conducted using pre-trained CLIP model with a ViT-B/16 backbone.

Table 4: **Detailed performance(%) of DeMul:** The detail top-1 accuracy score across 11 diverse downstream datasets, along with the average scores for each. '# Shot' represents the number of training samples per class used in few-shot learning.

| Method | # Shot | ImageNet | SUN397 | Stanford Cars | UCFI01 | Caltech101 | EuroSAT | FGVC Aircraft | Food101 | DTD | Oxford Flowers | Oxford Pets | Average |
|---|---|---|---|---|---|---|---|---|---|---|---|---|---|
| Zero-Shot CLIP | 0 | 68.4 | 62.3 | 65.1 | 64.8 | 92.3 | 48.8 | 24.8 | 84.7 | 44.6 | 70.4 | 88.4 | 65.0 |
| Zero-Shot dCLIP | 0 | 70.1 | 64.2 | 66.9 | 65.9 | 92.7 | 51.2 | 26.4 | 85.7 | 47.8 | 71.1 | 89.5 | 66.5 |
| Zero-Shot WaffleCLIP | 0 | 71.4 | 63.9 | 67.3 | 66.3 | 93.0 | 53.3 | 25.9 | 86.5 | 48.0 | 72.4 | 89.9 | 67.1 |
| Linear Probe CLIP | 1 | 32.1 | 41.6 | 35.7 | 53.7 | 79.9 | 49.2 | 19.6 | 43.0 | 34.6 | 69.7 | 44.1 | 45.8 |
|  | 2 | 44.9 | 53.7 | 50.3 | 65.8 | 89.0 | 62.0 | 26.4 | 61.5 | 40.8 | 85.1 | 58.4 | 58.0 |
|  | 4 | 54.9 | 63.0 | 63.4 | 73.3 | 92.1 | 77.1 | 32.3 | 73.2 | 55.7 | 92.0 | 71.2 | 68.0 |
|  | 8 | 62.2 | 69.1 | 73.7 | 79.3 | 93.4 | 84.4 | 39.4 | 79.8 | 63.5 | 96.1 | 78.4 | 74.5 |
|  | 16 | 67.3 | 73.3 | 80.4 | 82.1 | 95.4 | 87.2 | 45.4 | 82.9 | 69.9 | 97.4 | 85.3 | 78.8 |
| CoOp | 1 | 66.3 | 66.8 | 67.4 | 71.2 | 79.9 | 49.2 | 19.6 | 71.2 | 34.6 | 69.7 | 44.1 | 58.2 |
|  | 2 | 67.1 | 66.5 | 70.5 | 73.4 | 89.0 | 62.0 | 26.4 | 73.4 | 40.8 | 85.1 | 58.4 | 64.8 |
|  | 4 | 68.7 | 69.9 | 74.5 | 77.1 | 92.1 | 77.1 | 32.3 | 77.1 | 55.7 | 92.0 | 71.2 | 71.6 |
|  | 8 | 70.6 | 71.5 | 79.3 | 80.2 | 93.4 | 84.4 | 39.4 | 80.2 | 63.5 | 96.1 | 78.4 | 76.1 |
|  | 16 | 71.9 | 74.7 | 83.1 | 82.2 | 95.4 | 87.2 | 45.4 | 82.2 | 69.9 | 97.4 | 85.3 | 79.5 |
| MaPLe | 1 | 62.7 | 64.8 | 66.7 | 71.8 | 92.6 | 71.8 | 26.7 | 80.5 | 52.1 | 83.3 | 89.1 | 69.3 |
|  | 2 | 65.1 | 67.1 | 71.6 | 74.6 | 94.0 | 78.3 | 30.9 | 81.5 | 55.5 | 88.9 | 90.9 | 72.6 |
|  | 4 | 67.7 | 70.7 | 75.3 | 78.5 | 94.4 | 84.5 | 34.9 | 81.8 | 61.0 | 92.7 | 91.9 | 75.8 |
|  | 8 | 70.3 | 73.2 | 79.5 | 81.4 | 95.2 | 87.7 | 42.0 | 83.6 | 66.5 | 95.8 | 92.6 | 78.9 |
|  | 16 | 72.3 | 75.5 | 83.6 | 85.0 | 96.0 | 92.3 | 48.4 | 85.3 | 71.3 | 97.0 | 92.8 | 81.8 |
| PromptSRC | 1 | 68.1 | **69.7** | 69.4 | 74.8 | 93.7 | 73.1 | 27.7 | 84.9 | 56.2 | 85.9 | 92.0 | 72.3 |
|  | 2 | 69.8 | 71.6 | 73.4 | 78.5 | 94.5 | 79.4 | 31.7 | 85.7 | 60.0 | 91.2 | 92.5 | 75.3 |
|  | 4 | 71.1 | 74.0 | 77.1 | 81.6 | 95.3 | 86.3 | 37.5 | 86.1 | 65.5 | 93.9 | 93.4 | 78.3 |
|  | 8 | 72.3 | 75.7 | 81.0 | 84.3 | 95.7 | 88.8 | 43.3 | 86.9 | 69.9 | 96.3 | 93.5 | 80.7 |
|  | 16 | 73.2 | 77.2 | 83.8 | 86.5 | 96.1 | 92.4 | 47.1 | 87.5 | 72.7 | 97.6 | 93.7 | 82.9 |
| GalLoP | 1 | 70.8 | 68.3 | 71.0 | 74.5 | 94.7 | 79.5 | 32.8 | 85.8 | 56.0 | 84.9 | 92.2 | 73.7 |
|  | 2 | 71.5 | 71.6 | 76.1 | 78.2 | 95.1 | 78.6 | 36.2 | **86.5** | 61.3 | 92.5 | 93.2 | 76.4 |
|  | 4 | 72.2 | 73.8 | 78.3 | 81.9 | 95.5 | 82.7 | 44.1 | 86.9 | 66.9 | 95.2 | 93.4 | 79.2 |
|  | 8 | 73.8 | 75.7 | 84.9 | 85.0 | 96.0 | 86.9 | 51.0 | 87.5 | 72.4 | **98.5** | 93.4 | 82.3 |
|  | 16 | 75.1 | 77.2 | 89.2 | 86.9 | 96.7 | 90.1 | 58.3 | 87.9 | 75.5 | 97.9 | 94.1 | 84.4 |
| DeMul (Ours) | 1 | **72.3** | 68.5 | **74.7** | **76.9** | **95.1** | 80.7 | 34.3 | 86.1 | 62.1 | 87.2 | 93.0 | 75.5 |
|  | 2 | **72.6** | 73.1 | **78.5** | 80.0 | **95.4** | 83.1 | 40.2 | 86.4 | 67.9 | 93.4 | 93.6 | 78.6 |
|  | 4 | **73.1** | 74.8 | 81.3 | 83.2 | 96.0 | 87.4 | 47.1 | 87.0 | 70.3 | 96.0 | 94.0 | 80.9 |
|  | 8 | **74.3** | 76.4 | 87.7 | 85.7 | 96.4 | 89.7 | 54.3 | 87.5 | 75.1 | 97.9 | 94.3 | 83.6 |
|  | 16 | **76.1** | 77.9 | 89.9 | 87.7 | 96.9 | 92.5 | 59.9 | 87.8 | 77.0 | 98.0 | 94.9 | 85.3 |

