# OpenReview forum: "Weighted Multi-Prompt Learning with Description-free Large Language Model Distillation"
_ICLR.cc/2025/Conference — ICLR 2025 Poster_

### Official Review · Reviewer_oNaD · 2024-10-31

**Soundness:** 3
**Presentation:** 3
**Contribution:** 3
**Rating:** 6
**Confidence:** 4

**Summary:**

This paper introduces Description-free Multi-prompt Learning (DeMul), a new method for prompt learning in vision-language models that eliminates the need for descriptive text by directly distilling knowledge from large language models (LLMs) like GPT. The approach optimizes prompts in the LLM embedding space and incorporates weighted multi-prompt learning, allowing prompts to capture diverse semantics and improve few-shot classification performance. Experimental results across 11 datasets show that DeMul outperforms traditional description-based methods, simplifying prompt learning while preserving rich semantics.

**Strengths:**

1.The proposed method does not require text descriptions or specific prompt design, while still preserving rich semantic expression.
2.The proposed Weighted Multi-Prompt Learning effectively captures the relative importance of different prompts, enhancing model adaptability and performance.

**Weaknesses:**

1.Figure 1 highlights the variability in GPT-based descriptions, but this diversity can actually be beneficial in image classification. The goal of descriptions is to enrich the semantic variety within each class, helping the model handle diverse images within the same category, rather than strictly aligning with the class name. By dismissing this variability, the proposed approach might miss out on the robustness provided by diverse descriptions.

2.Mapping prompt descriptions to the GPT embedding space may reduce interpretability, as this space is abstract and lacks human-interpretable structure. The optimizations in this space might rely more on implicit semantic relationships rather than explicit, understandable features, which could obscure the interpretability of the learned representations.

3.The paper does not reference or explain Algorithm 1: Training process of DeMul, which leaves readers without guidance on understanding the step-by-step training procedure of the proposed method.

4.The description of Equation (4) suggests averaging classification probabilities across prompts, but the actual implementation seems to involve averaging the text embeddings of prompts instead.

5.The Preliminaries and Background section lacks appropriate citations, which limits the clarity.

6.The paper specifies fixed values for the regularization weight  and the loss balance parameter, but does not provide ablation studies to justify these choices.

7.The Related Work section lacks a clear articulation of the limitations in existing methods and how the proposed approach addresses these gaps.

8.The paper claims that the weights $\{w_{ij}\}$ for each class are normalized, ensuring that $\(\sum_{j=1}^{M} w_{ij} = 1\)$. However, in Figure 5, the sum of the weights for the five prompts appears to exceed 1, which contradicts this claim.

9.In the ablation study presented in Table 1, the difference in accuracy between "Ours w/o weighted" (85.2) and "Ours" (85.3) in the 16-shot setting is minimal, with only a 0.1% improvement when using the proposed weighting mechanism. This marginal difference raises questions about the practical impact of the weighted approach.

**Questions:**

No questions or suggestions.

---

> ### Author Response · Authors · 2024-11-24
> **Response for oNaD - Part 1**
>
> Upon reviewing the feedback, we noticed that the comments appear to be identical to those provided by the reviewer above(reviewer JO4M).  Since the content is exactly the same, we have provided the same response. If you have any information regarding this, we would greatly appreciate it if you could share them with us.
>
> **Response 1.**
> > Figure 1 highlights the variability in GPT-based descriptions, but this diversity can actually be beneficial in image classification. The goal of descriptions is to enrich the semantic variety within each class, helping the model handle diverse images within the same category, rather than strictly aligning with the class name. By dismissing this variability, the proposed approach might miss out on the robustness provided by diverse descriptions.
>
> As you pointed out, the goal of descriptions is to enrich the semantic variety within each class. However, one of the key limitations we observed in existing methods is that the diversity of descriptions is effectively "restricted" to a fixed set of predefined sentences. As shown in Figure 1, description-based methods heavily rely on descriptions extracted from the LLM at a single point in time. This reliance inherently limits the diversity of descriptions, regardless of whether the extracted descriptions actually contribute to semantic enrichment.
>
> In contrast, our proposed DeMul dynamically learns meaningful semantics during training, allowing it to reference a much broader distribution. This approach aligns with the design philosophy of learnable prompts, as first introduced in CoOp[a]. For example, CoOp demonstrated that trivial variations in sentences, such as adding a "." or "a," might not contribute positively and, in some cases, could even harm robustness. By not restricting such variations and instead dynamically learning semantics, CLIP was able to achieve greater semantic richness and robustness, as validated by subsequent research.
>
> Therefore, while we agree with your observation that dismissing semantic diversity may compromise robustness, DeMul addresses this concern by ensuring richer diversity and, as a result, enhances robustness.
>
> [a] Learning to Prompt for Vision-Language Models, IJCV 2022, CVPR 2022
>
> **Response 2.**
> > Mapping prompt descriptions to the GPT embedding space may reduce interpretability, as this space is abstract and lacks human-interpretable structure. The optimizations in this space might rely more on implicit semantic relationships rather than explicit, understandable features, which could obscure the interpretability of the learned representations.
>
> We agree with the reviewer’s observation that it is difficult to explicitly interpret the learned prompts, and we acknowledge this as one of the limitations of our method. However, this limitation is not unique to our approach; all CoOp-based learnable prompt methods have reported similar challenges.
> Our method builds upon this foundation by incorporating the implicit knowledge of LLMs to generate richer semantics, which significantly contribute to performance improvements. We believe this represents another vast and promising research direction in prompt learning. For instance, as shown in Table 4 of the CoOp paper, progress has already been made in studying interpretability within this domain.
>
> In light of your comment, we have added this point to the further research in conclusion sections of our paper.  The revised texts are highlighted in \\(\textcolor{violet}{\text{violet}}\\) color.
>
> **Response 3.**
> > The paper does not reference or explain Algorithm 1: Training process of DeMul, which leaves readers without guidance on understanding the step-by-step training procedure of the proposed method.
>
> Thank you very much for pointing out the part we missed. We understand that the lack of guidance might have made it difficult for readers to follow the algorithm. We have addressed this by adding the necessary details into the implementation details section to help understand the algorithm overall, even though step-by-step specifics are not fully included to maintain the readability of the paper. This is also highlighted in \textcolor{violet}{violet} color.

---

> ### Author Response · Authors · 2024-11-24
> **Response for oNaD - Part 2**
>
> **Response 4.**
> > The description of Equation (4) suggests averaging classification probabilities across prompts, but the actual implementation seems to involve averaging the text embeddings of prompts instead.
>
> We are not entirely sure which part of our code you believe is performing "averaging the text embeddings of prompts". If there are any remaining questions or clarifications needed after this comment, please feel free to point them out again, and we will gladly review them further.
>
> In our implementation, we calculate the classification loss (`cls_loss`) by first applying `self.criterion(output, label)`, which is implemented using `nn.CrossEntropyLoss()`. This inherently involves averaging over classification probabilities which aligns well with equation (4). Subsequently, additional loss computations are incorporated into the overall loss calculation.
>
> **Response 5.**
> > The Preliminaries and Background section lacks appropriate citations, which limits the clarity.
>
> It seems that the citations for CLIP and CoOp were missing in the preliminaries and background section.  We have added these citations to clarify the content. This is also highlighted in violet color.
>
> **Response 6.**
> > The paper specifies fixed values for the regularization weight and the loss balance parameter, but does not provide ablation studies to justify these choices.
>
> Thank you for your valuable insight. Since another reviewer has raised the same question, we will address it in the general comment section.
>
> **Response 7.**
> > The Related Work section lacks a clear articulation of the limitations in existing methods and how the proposed approach addresses these gaps.
>
> We faced challenges in fitting all the necessary graphs for the main results and additional experiments within the strict 10-page limit. After much consideration, we decided to avoid redundancy by not repeating the limitations of existing methods, which were already mentioned in the Introduction, in the Related Works section. Instead, we chose to focus the Related Works section on providing essential background, such as Preliminaries, and introducing the baselines for comparison, ensuring readers can understand the Introduction and the context of our work.
>
> While we believed that the limitations of existing methods and how DeMul addresses them were well integrated into other sections, we understand that including this discussion in the Related Works section could be helpful. If you still feel this addition would improve the paper, we would be happy to revise accordingly.
>
> **Response 8.**
> > The paper claims that the weights $w_{ij}$ for each class are normalized, ensuring that $\sum_{j=1}^M w_{ij} = 1$. However, in Figure 5, the sum of the weights for the five prompts appears to exceed 1, which contradicts this claim.
>
> We appreciate the reviewer’s observation and agree that the explanation for Figure 5 was insufficient, which may have caused confusion.
>
> As stated in the equation, we ensure that \\( \sum_{j=1}^M w_{ij} = 1 \\) during training. The top graph in Figure 5 represents the relative variations in prompt weights rather than their absolute values. Specifically, it shows how much each prompt's weight has changed from its initial value. A value greater than 1.00 indicates an increase in weight, while a value less than 1.00 indicates a decrease.
>
> This visualization was designed to emphasize the upward and downward trends in weights rather than their absolute values. When compared with the lower graph, the focus is on the dynamics of the weights rather than their sum. We have clarified this point by adding the phrase “by their relative changes from their initialized values” to the caption of Figure 5 to address any potential confusion.
> This is also highlighted in violet color.

---

> ### Author Response · Authors · 2024-11-24
> **Response for oNaD - Part 3**
>
> **Response 9.**
> > In the ablation study presented in Table 1, the difference in accuracy between "Ours w/o weighted" (85.2) and "Ours" (85.3) in the 16-shot setting is minimal
>
> We acknowledge that the accuracy improvement of 0.1\% in the 16-shot setting is indeed minimal. However, we believe it is insufficient to conclude that the weighting mechanism is ineffective based solely on this result.
>
> First, the core idea behind prompt weighting can be summarized as determining "how much useful semantics should be referenced during distillation." This implies a strong dependency between the weighting mechanism and distillation, which may lead to varying degrees of interaction depending on the experimental setup. While this interaction may be less apparent in the 16-shot setting, we observed consistent and significant contributions to performance improvements across other N-shot settings. This demonstrates the effectiveness of the combined approach rather than the mechanisms in isolation.
>
> Second, the introduction of the weighting mechanism is not solely aimed at improving accuracy but also enables the model to learn the relative importance of each prompt, contributing to more stable and generalizable results. In this way, the weighting mechanism plays a crucial role in optimizing both the training process and overall model efficiency.
>
> Third, few-shot learning is inherently more challenging as the amount of available data decreases. As previously mentioned, the dependency between variables may not scale linearly with N-shot. However, the improvements observed—0.3\% in the 1-shot setting and 0.8\% in the 2-shot setting—are substantially larger than in the 16-shot scenario, highlighting the critical role of the weighting mechanism in more constrained environments.

---

> > ### Comment · Reviewer_oNaD · 2024-11-24
> >
> > I appreciate your response, which addressed my concerns and clarified key points. Considering all feedback, I have updated my score to 6.

---

> > > ### Author Response · Authors · 2024-11-24
> > > **Thanks for the reply**
> > >
> > > Thank you as well for leaving such positive comments. Your feedback provided us with a valuable opportunity to reflect on areas of our paper that might have been lacking, we truly appreciate the thoughtful insights you have shared.
> > >
> > > Thank you once again for your invaluable contributions to the improvement of our paper.
> > >
> > > Sincerely,
> > > The authors

---

### Official Review · Reviewer_jo4m · 2024-10-31

**Soundness:** 3
**Presentation:** 3
**Contribution:** 3
**Rating:** 6
**Confidence:** 5

**Summary:**

This paper proposes Description-free Multi-prompt Learning (DeMul), a method that enhances image recognition tasks by distilling knowledge directly from LLMs without needing descriptive text prompts. This method enables flexible prompt learning by mapping learnable prompts into the LLM embedding space and using weighted multi-prompt learning to adjust prompt importance dynamically. Experimental results across 11 datasets show that DeMul outperforms conventional description-based methods and other prompt optimization approaches.

**Strengths:**

(1) Well-structured and clearly explains the limitations of existing description-based prompt methods.

(2) Extensive experiments across 11 datasets demonstrate the robustness and effectiveness of the proposed method.

(3) While the novelty in the method may be incremental, applying weighted multi-prompt learning without description reliance addresses the practical limitations of prior work.

**Weaknesses:**

1. Figure 1 highlights the variability in GPT-based descriptions, but this diversity can actually be beneficial in image classification. The goal of descriptions is to enrich the semantic variety within each class, helping the model handle diverse images within the same category, rather than strictly aligning with the class name. By dismissing this variability, the proposed approach might miss out on the robustness provided by diverse descriptions.

2. Mapping prompt descriptions to the GPT embedding space may reduce interpretability, as this space is abstract and lacks human-interpretable structure. The optimizations in this space might rely more on implicit semantic relationships rather than explicit, understandable features, which could obscure the interpretability of the learned representations.

3.The paper does not reference or explain Algorithm 1: Training process of DeMul, which leaves readers without guidance on understanding the step-by-step training procedure of the proposed method.

4. The description of Equation (4) suggests averaging classification probabilities across prompts, but the actual implementation seems to involve averaging the text embeddings of prompts instead.

5.T he Preliminaries and Background section lacks appropriate citations, which limits the clarity.

6. The paper specifies fixed values for the regularization weight  and the loss balance parameter, but does not provide ablation studies to justify these choices.

7. The Related Work section lacks a clear articulation of the limitations in existing methods and how the proposed approach addresses these gaps.

8.The paper claims that the weights $\{w_{ij}\}$ for each class are normalized, ensuring that $\(\sum_{j=1}^{M} w_{ij} = 1\)$. However, in Figure 5, the sum of the weights for the five prompts appears to exceed 1, which contradicts this claim.

9. In the ablation study presented in Table 1, the difference in accuracy between "Ours w/o weighted" (85.2) and "Ours" (85.3) in the 16-shot setting is minimal

**Questions:**

See the weaknesses.

---

> ### Author Response · Authors · 2024-11-24
> **Response for jo4m - Part 1**
>
> Thank you for providing diverse perspectives and valuable feedback, which have greatly helped us improve our paper.  Below, we have provided our responses or additional experimental results for the nine comments you raised. We sincerely hope that our responses address your questions and concerns. If there are any further points of discussion after this, we would be more than happy to continue the conversation.
>
> **Response 1.**
> > Figure 1 highlights the variability in GPT-based descriptions, but this diversity can actually be beneficial in image classification. The goal of descriptions is to enrich the semantic variety within each class, helping the model handle diverse images within the same category, rather than strictly aligning with the class name. By dismissing this variability, the proposed approach might miss out on the robustness provided by diverse descriptions.
>
> As you pointed out, the goal of descriptions is to enrich the semantic variety within each class. However, one of the key limitations we observed in existing methods is that the diversity of descriptions is effectively "restricted" to a fixed set of predefined sentences. As shown in Figure 1, description-based methods heavily rely on descriptions extracted from the LLM at a single point in time. This reliance inherently limits the diversity of descriptions, regardless of whether the extracted descriptions actually contribute to semantic enrichment.
>
> In contrast, our proposed DeMul dynamically learns meaningful semantics during training, allowing it to reference a much broader distribution. This approach aligns with the design philosophy of learnable prompts, as first introduced in CoOp[a]. For example, CoOp demonstrated that trivial variations in sentences, such as adding a "." or "a," might not contribute positively and, in some cases, could even harm robustness. By not restricting such variations and instead dynamically learning semantics, CLIP was able to achieve greater semantic richness and robustness, as validated by subsequent research.
>
> Therefore, while we agree with your observation that dismissing semantic diversity may compromise robustness, DeMul addresses this concern by ensuring richer diversity and, as a result, enhances robustness.
>
> [a] Learning to Prompt for Vision-Language Models, IJCV 2022, CVPR 2022
>
> **Response 2.**
> > Mapping prompt descriptions to the GPT embedding space may reduce interpretability, as this space is abstract and lacks human-interpretable structure. The optimizations in this space might rely more on implicit semantic relationships rather than explicit, understandable features, which could obscure the interpretability of the learned representations.
>
> We agree with the reviewer’s observation that it is difficult to explicitly interpret the learned prompts, and we acknowledge this as one of the limitations of our method. However, this limitation is not unique to our approach; all CoOp-based learnable prompt methods have reported similar challenges.
> Our method builds upon this foundation by incorporating the implicit knowledge of LLMs to generate richer semantics, which significantly contribute to performance improvements. We believe this represents another vast and promising research direction in prompt learning. For instance, as shown in Table 4 of the CoOp paper, progress has already been made in studying interpretability within this domain.
>
> In light of your comment, we have added this point to the further research in conclusion sections of our paper.  The revised texts are highlighted in \\(\textcolor{violet}{\text{violet}}\\) color.
>
> **Response 3.**
> > The paper does not reference or explain Algorithm 1: Training process of DeMul, which leaves readers without guidance on understanding the step-by-step training procedure of the proposed method.
>
> Thank you very much for pointing out the part we missed. We understand that the lack of guidance might have made it difficult for readers to follow the algorithm. We have addressed this by adding the necessary details into the implementation details section to help understand the algorithm overall, even though step-by-step specifics are not fully included to maintain the readability of the paper. This is also highlighted in \textcolor{violet}{violet} color.

---

> ### Author Response · Authors · 2024-11-24
> **Response for jo4m - Part 2**
>
> **Response 4.**
> > The description of Equation (4) suggests averaging classification probabilities across prompts, but the actual implementation seems to involve averaging the text embeddings of prompts instead.
>
> We are not entirely sure which part of our code you believe is performing "averaging the text embeddings of prompts". If there are any remaining questions or clarifications needed after this comment, please feel free to point them out again, and we will gladly review them further.
>
> In our implementation, we calculate the classification loss (`cls_loss`) by first applying `self.criterion(output, label)`, which is implemented using `nn.CrossEntropyLoss()`. This inherently involves averaging over classification probabilities which aligns well with equation (4). Subsequently, additional loss computations are incorporated into the overall loss calculation.
>
> **Response 5.**
> > The Preliminaries and Background section lacks appropriate citations, which limits the clarity.
>
> It seems that the citations for CLIP and CoOp were missing in the preliminaries and background section.  We have added these citations to clarify the content. This is also highlighted in violet color.
>
> **Response 6.**
> > The paper specifies fixed values for the regularization weight and the loss balance parameter, but does not provide ablation studies to justify these choices.
>
> Thank you for your valuable insight. Since another reviewer has raised the same question, we will address it in the general comment section.
>
> **Response 7.**
> > The Related Work section lacks a clear articulation of the limitations in existing methods and how the proposed approach addresses these gaps.
>
> We faced challenges in fitting all the necessary graphs for the main results and additional experiments within the strict 10-page limit. After much consideration, we decided to avoid redundancy by not repeating the limitations of existing methods, which were already mentioned in the Introduction, in the Related Works section. Instead, we chose to focus the Related Works section on providing essential background, such as Preliminaries, and introducing the baselines for comparison, ensuring readers can understand the Introduction and the context of our work.
>
> While we believed that the limitations of existing methods and how DeMul addresses them were well integrated into other sections, we understand that including this discussion in the Related Works section could be helpful. If you still feel this addition would improve the paper, we would be happy to revise accordingly.
>
> **Response 8.**
> > The paper claims that the weights $w_{ij}$ for each class are normalized, ensuring that $\sum_{j=1}^M w_{ij} = 1$. However, in Figure 5, the sum of the weights for the five prompts appears to exceed 1, which contradicts this claim.
>
> We appreciate the reviewer’s observation and agree that the explanation for Figure 5 was insufficient, which may have caused confusion.
>
> As stated in the equation, we ensure that \\( \sum_{j=1}^M w_{ij} = 1 \\) during training. The top graph in Figure 5 represents the relative variations in prompt weights rather than their absolute values. Specifically, it shows how much each prompt's weight has changed from its initial value. A value greater than 1.00 indicates an increase in weight, while a value less than 1.00 indicates a decrease.
>
> This visualization was designed to emphasize the upward and downward trends in weights rather than their absolute values. When compared with the lower graph, the focus is on the dynamics of the weights rather than their sum. We have clarified this point by adding the phrase “by their relative changes from their initialized values” to the caption of Figure 5 to address any potential confusion.
> This is also highlighted in violet color.

---

> ### Author Response · Authors · 2024-11-24
> **Response for jo4m - Part 3**
>
> **Response 9.**
> > In the ablation study presented in Table 1, the difference in accuracy between "Ours w/o weighted" (85.2) and "Ours" (85.3) in the 16-shot setting is minimal
>
> We acknowledge that the accuracy improvement of 0.1\% in the 16-shot setting is indeed minimal. However, we believe it is insufficient to conclude that the weighting mechanism is ineffective based solely on this result.
>
> First, the core idea behind prompt weighting can be summarized as determining "how much useful semantics should be referenced during distillation." This implies a strong dependency between the weighting mechanism and distillation, which may lead to varying degrees of interaction depending on the experimental setup. While this interaction may be less apparent in the 16-shot setting, we observed consistent and significant contributions to performance improvements across other N-shot settings. This demonstrates the effectiveness of the combined approach rather than the mechanisms in isolation.
>
> Second, the introduction of the weighting mechanism is not solely aimed at improving accuracy but also enables the model to learn the relative importance of each prompt, contributing to more stable and generalizable results. In this way, the weighting mechanism plays a crucial role in optimizing both the training process and overall model efficiency.
>
> Third, few-shot learning is inherently more challenging as the amount of available data decreases. As previously mentioned, the dependency between variables may not scale linearly with N-shot. However, the improvements observed—0.3\% in the 1-shot setting and 0.8\% in the 2-shot setting—are substantially larger than in the 16-shot scenario, highlighting the critical role of the weighting mechanism in more constrained environments.

---

> > ### Comment · Reviewer_jo4m · 2024-11-24
> > **Thanks for your response.**
> >
> > Thank you for your response. It addressed all my questions, including some misunderstandings. After reviewing the feedback from other reviewers as well, I am happy to increase my score as 6.

---

> > > ### Author Response · Authors · 2024-11-24
> > > **Thanks for the reply**
> > >
> > > We are delighted to hear that our additional efforts have addressed your questions and resolved any misunderstandings. We sincerely appreciate your positive feedback and are deeply grateful for your kind consideration in raising the score as well.
> > >
> > > Thank you once again for your invaluable contributions to the improvement of our paper.
> > >
> > > Sincerely,
> > > The authors

---

### Official Review · Reviewer_3ZUA · 2024-11-02

**Soundness:** 3
**Presentation:** 3
**Contribution:** 3
**Rating:** 8
**Confidence:** 3

**Summary:**

This paper proposes a novel approach called DeMul for few-shot image recognition. The method leverages pre-trained VLMs and bypasses the need for extracting textual descriptions from LLMs by directly distilling knowledge from the LLM into learnable prompts. This description-free approach effectively addresses the variability and potential unreliability of LLM-generated descriptions in previous methods.

**Strengths:**

Originality: The description-free distillation is a novel concept that cleverly addresses a key limitation of existing LLM-enhanced prompt learning methods. The weighted multi-prompt learning strategy further enhances the originality of the approach.

Quality: The paper presents compelling empirical results across 11 diverse datasets, with DeMul consistently outperforming existing zero-shot and few-shot methods, including state-of-the-art techniques like GalLoP. The comprehensive experimental setup and ablation studies further strengthen the quality of the evaluation.

Clarity: The paper is well-organized and easy to understand. The motivation, methodology, and experimental results are presented clearly and concisely. The figures and tables are informative and effectively illustrate the key concepts and findings.

Significance: DeMul simplifies the prompt engineering process and improves the reliability of VLM-based image recognition by eliminating the need for textual descriptions. The significant performance gains in few-shot learning scenarios demonstrate its practical value. Furthermore, its core ideas could potentially be applied to other VLM tasks, broadening its impact.

**Weaknesses:**

Limited analysis of the mapping function: While the paper introduces the concept of a mapping function, the exploration of its properties remains superficial, lacking quantitative analysis to assess its effectiveness.

Dependence on training data distribution: The paper acknowledges the potential sensitivity of DeMul to the training data distribution but lacks an in-depth investigation of this sensitivity.

Shared learnable vectors: While improving memory efficiency, shared learnable vectors might limit the model's ability to capture class-specific prompt information.

Lack of theoretical support: The paper primarily focuses on empirical results and lacks a strong theoretical foundation to explain the effectiveness of the method.

**Questions:**

1、Provide a more detailed analysis of the mapping function's properties and explore alternative mapping strategies.
2、Investigate the performance variations under different training data splits and explore data augmentation techniques to improve robustness.
3、Evaluate the memory savings from shared learnable vectors and explore alternative approaches to balance memory efficiency and representational power.
4、Attempt to provide some theoretical analysis to explain the observed performance gains.

---

> ### Author Response · Authors · 2024-11-24
> **Response for 3ZUA - Part 1**
>
> We sincerely appreciate your positive feedback on our paper and the valuable insights you’ve shared, which will help us further improve our work. Within the short rebuttal period, we have done our best to address the points raised and incorporate any feasible updates. We hope our responses below address your comments, and we would be happy to engage in further discussion if you have any additional questions or suggestions for improvement.
>
> **Response 1.**
> > Limited analysis of the mapping function: While the paper introduces the concept of a mapping function, the exploration of its properties remains superficial, lacking quantitative analysis to assess its effectiveness.
> (Matching with Q1) Provide a more detailed analysis of the mapping function's properties and explore alternative mapping strategies.
>
> We appreciate the reviewer’s insightful comments regarding the computational cost, dataset size, and potential noise or biases in our approach. To address these concerns, we conducted the following detailed analyses and experiments:
>
> **(1) Computational Cost**
> The mapping function is designed to be lightweight and computationally efficient. Below are the specific details:
> - Model architecture: 5-layered MLP
> - Model size: 0.72M parameters (combined for \\( \psi \\) and \\( \varphi \\))
> - Input/Output dimensions: CLIP embedding dim: 512, GPT embedding dim: 1536
> - Training time: 30 minutes for 100 epochs on 1 NVIDIA V100 GPU (batch size = 32)
> - Average inference time per sample: 0.35 ms
> - Memory usage: Maximum: 431.58 MB, Average: 431.56 MB
>
> These results confirm that the mapping function incurs minimal computational overhead during both pretraining and inference.
>
> **(2) Mapping Accuracy**
> To evaluate the mapping accuracy, we split the dataset \\( \mathcal{D}_{\text{mapping}} \\) into an 8:2 train:test ratio. Using a model pretrained for 100 epochs, we assessed the cosine similarity between embeddings. The results are shown in the table below, where \\( \varphi(t_i) \\) represents the embeddings mapped to the GPT space, and \\( \psi(\varphi(t_i)) \\) represents the embeddings cycled back to the CLIP space. The high accuracy demonstrates that the mapping function preserves key semantics critical for distillation.
>
> **(3) Dataset Size**
> To further investigate the robustness of the mapping function, we reduced the size of \\( \mathcal{D}_{\text{mapping}} \\) by randomly sampling 50% of the dataset. The table above shows that even with 50% of the data, the mapping function achieves comparable accuracy. This indicates that lightweight datasets or diverse distributions could be equally effective, suggesting that our approach is not overly reliant on dataset size.
>
> **Train and Test Accuracy(%)**
> |        | Train (\\( \varphi(t_i) \\)) | Train (\\( \psi(\varphi(t_i)) \\)) | Test (\\( \varphi(t_i) \\)) | Test (\\( \psi(\varphi(t_i)) \\)) |
> |--------|-------------------------------|-------------------------------------|------------------------------|------------------------------------|
> | 100%   | 99.27                         | 97.78                              | 96.25                        | 95.11                             |
> | 50%    | 98.60                         | 95.91                              | 95.09                        | 93.85                             |
>
> **(4) Comparison with CyclicGAN**
> In response to the suggestion to explore alternative mapping strategies, we conducted an experiment using CyclicGAN. While CyclicGAN did not perform as well as our method, possibly due to incomplete optimization within the available timeframe, this experiment highlights the efficacy of our approach. Below are the results and details for CyclicGAN:
> - Model size: 7.35M parameters (combined for \\( \psi \\) and \\( \varphi \\))
> - Training time: 80 minutes for 100 epochs on 1 NVIDIA V100 GPU (batch size = 32)
> - Average memory usage: 582.59 MB
>
> **Train and Test Accuracy(%)**
> |        | Train (\\( \varphi(t_i) \\)) | Train (\\( \psi(\varphi(t_i)) \\)) | Test (\\( \varphi(t_i) \\)) | Test (\\( \psi(\varphi(t_i)) \\)) |
> |--------|-------------------------------|-------------------------------------|------------------------------|------------------------------------|
> | 100%   | 89.34                         | 82.10                              | 83.29                        | 77.28                             |
>
> **Response 2.**
> > Dependence on training data distribution: The paper acknowledges the potential sensitivity of DeMul to the training data distribution but lacks an in-depth investigation of this sensitivity.
> (Matching with Q2) Investigate the performance variations under different training data splits and explore data augmentation techniques to improve robustness.
>
> Thank you for your valuable insight. Since another reviewer has raised the same question, we will address it in the general comment section.

---

> ### Author Response · Authors · 2024-11-24
> **Response for 3ZUA - Part 2**
>
> **Response 3.**
> > Shared learnable vectors: While improving memory efficiency, shared learnable vectors might limit the model's ability to capture class-specific prompt information.
> (Matching with Q3) Evaluate the memory savings from shared learnable vectors and explore alternative approaches to balance memory efficiency and representational power.
>
> We completely agree with the reviewer’s observation. As mentioned in the limitation section of our conclusion, we acknowledge the trade-off between memory efficiency and the ability to capture class-specific prompts.
>
> Currently, (1) we have focused on demonstrating the goodness of distillation itself, aligning with existing works, and (2) have not incorporated approaches aimed at resolving memory efficiency constraints, which is why we opted for shared learnable vectors.
> However, in our ongoing future work, we are actively exploring ways to address this trade-off and investigating the potential benefits of employing class-specific prompts.
>
> **Response 4.**
> > Lack of theoretical support: The paper primarily focuses on empirical results and lacks a strong theoretical foundation to explain the effectiveness of the method.
> (Matching with Q4) Attempt to provide some theoretical analysis to explain the observed performance gains.
>
> We strongly agree that our work lacks a theoretical basis. We did not consider the need for a theoretical background, as we believed that the novelty of our work lies in the distillation of LLM representations and that the experimental results, which positively correlate with LLM model performance, offer sufficient support. As an additional experiment to explain why Cyclic mapping works so well, we observed the angle and size properties of the pre-trained CLIP text encoder. Although this may not fully address the question, it provides some insights.
> In the future, we would like to investigate whether the text encoders of other VLMs share the characteristics we observed above and explore whether these properties can be mathematically verified.
>
> For the pre-trained text encoder \\( g \\) of CLIP, we sampled \\( x_i \sim \mathcal{N}(0, 0.02) \\) for \\( 1 \leq i \leq 1000 \\).
>
> **Experiment 1:**
>   - Computed sequences \\(\\{g^{(j)}(x_i)\\}_{j=0}^{40}\\) for \\(1 \\leq i \\leq 1000\\).
>   - Observed that each sequence \\( \\{ ||\\{g^{(j)}(x_i)||\_2\\}_{j=0}^{40} \\) is monotonically increasing.
>   - Found that when \\(\|x\|_2 < 20\\), \\(\cos\angle(x, g(x)) \leq 0.2\\), and when \\(||x||_2 > 100\\), \\(\cos\angle(x, g(x)) \approx 1\\).
>
>   From this experiment, we observed that \\(g\\) preserves the angle of a point in \\(\mathbb{R}^{512}\setminus B_R(0)\\) for some \\(R > 0\\).
>
>   **Experiment 2:**
>   - Selected a dummy class \\(c \in \mathbb{R}^{512}\setminus B_{100}(0)\\).
>   - Concatenated \\(c\\) to samples to create \\(c \oplus x_i \sim \mathcal{N}(c \oplus 0, 0.02)\\) for \\(1 \leq i \leq 1000\\).
>   - Computed \\(\\{g(c \oplus x_i)\\}_{i=1}^{1000}\\) and visualized its 2D PCA projection.
>
> From this experiment, we observed that the variance of \\(\\{g(c \oplus x_i)\\}_{i=1}^{1000}\\)  is larger than. \\( \\{c \oplus x_i\\}\_{\{i=1\}}^{1000} \\) .
>
> From the above two experiments, we conclude that the scaling property of \\( g \\) and its angle-preserving property for vectors beyond a certain magnitude make it more reasonable to perform classification based on cosine similarity than Euclidean distance.
> Furthermore, the angle-preserving property of cyclic mapping makes it a well-compatible technique for LLM distillation using cosine similarity.
>
> In future research, we will explore whether the text encoders of other VLMs exhibit these two properties and investigate whether these properties can be mathematically verified.

---

### Official Review · Reviewer_eZLz · 2024-11-03

**Soundness:** 3
**Presentation:** 3
**Contribution:** 2
**Rating:** 6
**Confidence:** 5

**Summary:**

The paper proposes to learn richer prompts for Vision Language Models (VLMs) by mapping them to the Embedding Endpoint of GPTs. They aim to capture meaningful semantics by directly distilling the pre-trained knowledge of GPT without the use of explicit description queries. The improvement in the quality of prompts is achieved by minimizing the distillation loss between the prompt embedding space and the LLM embedding space to better capture the class orientations. The authors have also used a multi-prompt setting to assess the best semantics from a weighted prompt pool. The authors use a multi-prompt setting with prompt weighting. The approach is shown to work well in few-shot image classification scenario.

**Strengths:**

1.	The paper looks into an interesting direction of enriching prompts using cyclic distillation from LLM embedding spaces. It mitigates the explicit description queries for the class-names of a given dataset.
2.	The description-free approach with multi-prompt weighting seems to be effective in few-shot image classification.
3.	The paper is well organized and easy to follow.

**Weaknesses:**

1.	It seems that the pretraining has to be done on a dataset which covers all the category names in the evaluation. This would significantly limit the generalizability of the method. Also is class-names a good attribute for mapping the LLM embeddings?
2.	As mapping functions need to be pretrained, what is the computational cost incurred? Also, it seems that the comprehensive dataset required a lot of samples from a long list of class-names to cover all datasets. Such an approach is prone to noise and biases. The authors should clarify the function pretraining in detail and make their approach more generalizable.
3.	Considering Figure 3, the performance of DeMul is negligible in 6 out of 11 datasets compared to other approaches such as GalLoP. This significantly diminishes the efficacy of the approach.
4.	The major weakness of the paper is very limited evaluation. The authors have limited their approach to only few-shot image classification while all the recent relevant works [CoCoOp (Zhou et al., 2022a), GalLoP (Lafon et al., 2024), MaPLe (Khattak et al., 2023a)] show their efficacy on 1. Domain Generalization 2. Base-to-New Generalization and 3. Cross-Dataset Transfer in addition to few-shot downstream image classification. Without such comparison it is hard to judge the goodness of the proposed approach beyond the single task of few-shot downstream image classification.
5.	Distilling LLM text to learnable prompts and prompt weighting separately have been explored in some of the recent works. Such as [a] or [b]. The author should discuss these references in the related work section and explicitly explain how their work is distinct from these similar studies.

[a] PromptKD: Unsupervised Prompt Distillation for Vision-Language Models, CVPR 2024

[b] A Simple Zero-shot Prompt Weighting Technique to Improve Prompt Ensembling in Text-Image Models, ICML 2023

**Questions:**

The weaknesses are already given in the question form. In addition, here are a few additional questions.
1. How is the value of $\alpha$ chosen? Is there any sensitivity analysis on $\alpha$?
2. Are the experiments run for different random splits of training data? Then are the reported values the average over these?

---

> ### Author Response · Authors · 2024-11-24
> **Response for eZLz - Part 1**
>
> Thank you for your thorough analysis and constructive feedback on our paper. Here are our responses to the four weaknesses and one question you pointed out. We hope these answers address your concerns to some extent, and we would be happy to engage in further discussions if you have additional questions or comments.
>
> **Response 1.**
> > It seems that the pretraining has to be done on a dataset which covers all the category names in the evaluation. This would significantly limit the generalizability of the method. Also is class-names a good attribute for mapping the LLM embeddings?
>
> We would like to clarify that pretraining on a dataset containing all the category names was only performed to align the embedding spaces of CLIP and GPT through the mapping function. The CLIP model and prompts, which directly affect the final performance, were not exposed to or reliant on the evaluation dataset during this process. The mapping function’s sole purpose is to ensure alignment between the two embedding spaces. It does not involve learning the semantic information of category names. As long as alignment is guaranteed, any other dataset could be used, even without the specific category names. Therefore, we believe this process does not compromise the generalizability of our method.
>
> Additionally, class names are more than simple tags; they encapsulate rich semantic information, such as shape, behavior, and context. Prior work[c] has demonstrated that utilizing class names instead of predefined class indices enhances performance in image recognition tasks.
>
> [c] What’s in a Name? Beyond Class Indices for Image Recognition
>
> **Response 2.**
> >  As mapping functions need to be pretrained, what is the computational cost incurred? Also, it seems that the comprehensive dataset required a lot of samples from a long list of class-names to cover all datasets. Such an approach is prone to noise and biases. The authors should clarify the function pretraining in detail and make their approach more generalizable.
>
> Thank you for highlighting these concerns regarding computational cost, dataset dependency, and potential noise or biases in the pretraining process. Below, we address these points with experimental results and analysis.
>
> **Computational Cost and Model Details**
> The mapping function is lightweight and designed to minimize pretraining overhead, ensuring that the approach is computationally efficient and scalable. Here are the specific details:
> - Model architecture: 5-layered MLP
> - Model size: 0.72M parameters (\\( \psi + \varphi \\))
> - Input/Output dimensions: CLIP embedding dim: 512, LLM embedding dim: 1536
> - Training time: 30 minutes for 100 epochs on 1 NVIDIA V100 GPU (batch size = 32)
> - Average inference time per sample: 0.35 ms
> - Memory usage: Maximum: 431.58 MB, Average: 431.56 MB
>
> These metrics demonstrate that the mapping function introduces minimal computational overhead, making it feasible for practical deployment.
>
> **Addressing Noise and Bias Concerns**
> To evaluate the robustness of the mapping function against noise and biases, we conducted additional experiments using reduced datasets. Specifically, we randomly sampled 50% of \\( \\mathcal{D}\_{\text{mapping}} \\) and compared the performance with the full dataset. The table below shows the accuracy for train and test splits (8:2 in \\( D_{\text{mapping}} \\)), where \\( \varphi(t_i) \\) represents embeddings mapped to the GPT space, and \\( \psi(\varphi(t_i)) \\) represents embeddings cycled back to the CLIP space.
>
> **Train and Test Accuracy(%)**
> |        | Train (\\( \varphi(t_i) \\)) | Train (\\( \psi(\varphi(t_i)) \\)) | Test (\\( \varphi(t_i) \\)) | Test (\\( \psi(\varphi(t_i)) \\)) |
> |--------|----------------------------|----------------------------------|---------------------------|--------------------------------|
> | 100%   | 99.27                      | 97.78                           | 96.25                     | 95.11                         |
> | 50%    | 98.60                      | 95.91                           | 95.09                     | 93.85                         |
>
> These results show that reducing the dataset size by 50% has a negligible impact on mapping accuracy, highlighting the robustness of the mapping function against dataset variations. This also indicates that noise or biases in the dataset do not significantly affect the overall performance, alleviating concerns about over-dependence on a specific dataset.

---

> ### Author Response · Authors · 2024-11-24
> **Response for eZLz - Part 2**
>
> **Response 3.**
> > Considering Figure 3, the performance of DeMul is negligible in 6 out of 11 datasets compared to other approaches such as GalLoP. This significantly diminishes the efficacy of the approach.
>
> We understand that the performance gap achieved by DeMul might appear modest when observing the figure alone.
> However, CLIP is already a highly powerful model, with numerous prompt learning methods developed for it. In some datasets, CLIP has surpassed 90% accuracy, making further improvement challenging but critical for robust applications.
> DeMul demonstrates its effectiveness by achieving notable gains even in high-accuracy datasets, while also showing substantial improvements in datasets with lower initial accuracy. These consistent enhancements and the overall average performance boost validate the efficacy of our approach.
>
> Also, while we focused solely on demonstrating the effectiveness of LLM knowledge distillation and prompt weighting, we believe our method still holds necessity because it addresses different aspects compared to approaches like GalLoP and others. Depending on the scenario, the most suitable methods may vary, or combining multiple approaches could lead to additional performance improvements.
>
> **Response 4.**
> > The major weakness of the paper is very limited evaluation. The authors have limited their approach to only few-shot image classification while all the recent relevant works [CoCoOp (Zhou et al., 2022a), GalLoP (Lafon et al., 2024), MaPLe (Khattak et al., 2023a)] show their efficacy on 1. Domain Generalization 2. Base-to-New Generalization and 3. Cross-Dataset Transfer in addition to few-shot downstream image classification. Without such comparison it is hard to judge the goodness of the proposed approach beyond the single task of few-shot downstream image classification.
>
> We conducted additional experiments on two downstream tasks suggested in GalLoP[d], which we believe to be the most relevant baseline for our work. The experiments followed the same setup as GalLoP, using a model trained on ImageNet in the 16-shot setting and evaluating across four datasets to report the average results.
>
> **(1) Domain Generalization (Source: ImageNet, Target: ImageNet-V2, ImageNet-S, ImageNet-A, ImageNet-R)**
>
> | Method       | CLIP | CoOp | MaPLe | PromptSRC | ProDA | GalLoP | **DeMul (Ours)** |
> |--------------|------|------|-------|-----------|-------|--------|------------------|
> | Top-1 \\( \uparrow \\) | 57.2 | 59.4 | 60.3  | 60.7      | 60.0  | 61.3   | **62.1**        |
>
> **(2) OOD Detection (Source: ImageNet, Target: iNaturalist, SUN397, Places365, DTD)**
>
> | Method       | PLOT | PromptSRC | ProDA | CoOp | GalLoP | **DeMul (Ours)** |
> |--------------|------|-----------|-------|------|--------|------------------|
> | FPR95 \\( \downarrow \\) | 31.8 | 33.7      | 39.2  | 30.7 | 27.3   | **26.9**        |
>
> While we were unable to evaluate all three tasks mentioned within the given timeframe, we conducted these two tasks to demonstrate the effectiveness of DeMul in downstream tasks. We hope these experiments address your concerns and provide valuable insights into our approach.
>
> [d] GalLoP: Learning Global and Local Prompts for Vision-Language Models, ECCV 2024

---

> ### Author Response · Authors · 2024-11-24
> **Response for eZLz - Part 3**
>
> **Response 5.**
> > Distilling LLM text to learnable prompts and prompt weighting separately have been explored in some of the recent works. Such as [a] or [b]. The author should discuss these references in the related work section and explicitly explain how their work is distinct from these similar studies.
>
> We appreciate the reviewer’s suggestion regarding [a] PromptKD and [b] Zero-shot Prompt Weighting, as both works are indeed relevant to our research. However, there are clear distinctions in approach and contributions between these studies and ours.
>
> First, [a] PromptKD leverages teacher-student distillation as an unsupervised learning-based method to enhance the performance of VLMs such as CLIP.
> (1) This approach requires pretraining a teacher model like CLIP, which involves significant computational cost and dependency on the training data. In contrast, our method only requires lightweight mapping function pretraining, without the need to train an additional teacher model.
> (2) Furthermore, the performance of [a] is inherently constrained by the training data and domain of the teacher model. On the other hand, our method utilizes the powerful and broad semantic knowledge of LLMs, as demonstrated in many prior works, to distill a richer and more flexible set of textual representations, overcoming limitations imposed by training data.
>
> Second, [b] Zero-shot Prompt Weighting proposes a method to assign weights to prompts during inference by evaluating their importance.
> (1) This approach operates on a predefined set of prompts and a fixed model, without adapting to new tasks or optimizing prompts during training. In contrast, our method dynamically adjusts prompt weights during training, allowing the importance of prompts to be reflected in the learning process. As the weights directly influence the loss function during training, the learned prompts encode task- and dataset-specific semantic information more effectively. Interestingly, [b] underscores the necessity of considering the semantic information embedded in prompts, which further validates the need for DeMul’s dynamic prompt weighting mechanism.
>
> [a] PromptKD: Unsupervised Prompt Distillation for Vision-Language Models, CVPR 2024
> [b] A Simple Zero-shot Prompt Weighting Technique to Improve Prompt Ensembling in Text-Image Models, ICML 2023
>
> **Response for Questions-1. and Questions-2.**
> Thank you for your valuable insight. Since another reviewer has raised the same question, we will address it in the general comment section.

---

> > ### Comment · Reviewer_eZLz · 2024-11-24
> > **Thank you for the responses**
> >
> > Had a detailed read of the responses to my questions as well as those to my fellow reviewers' questions. I felt the authors did a good job at addressing some of the common questions (sensitivity analysis on hyperparameters, dependence or not on the pretraining dataset) and question specific mine (nefligable performance improvement on majority pf the evaluation datasets, lack of experiments related to OOD or Domain Generalizatio) and my fellow reviewers (Lack of theoretical support). As most of the queries are resolved, I am happy to increase the rating.

---

> > > ### Author Response · Authors · 2024-11-24
> > > **Thanks for the reply**
> > >
> > > We are delighted to hear that our response has been helpful. We deeply appreciate your thorough reading and understanding of our detailed reply, as well as your thoughtful engagement with the comments from fellow reviewers. Your kind words of recognition and encouragement mean a great deal to us. Furthermore, we would be truly grateful if you could reflect your positive impression in your score.
> > >
> > > Thank you once again for your invaluable contributions to the improvement of our paper.
> > >
> > > Sincerely,
> > > The authors

---

### Official Review · Reviewer_RgBn · 2024-11-04

**Soundness:** 3
**Presentation:** 3
**Contribution:** 3
**Rating:** 6
**Confidence:** 5

**Summary:**

This paper explores using large language models (LLMs) to enhance prompt learning for vision-language models. It proposes a description-free method to distill knowledge from LLMs into learnable prompts by directly aligning their embeddings. Additionally, a multi-prompt strategy is examined to improve performance. Experiments on few-shot learning benchmarks and ablation studies demonstrate the effectiveness of the proposed method.

**Strengths:**

1. The paper is well-crafted with clear expressions, making it easy to understand.

2. The description-free distillation approach for training learnable prompts is interesting, novel, and effective. It is also more cost-effective than description-needed methods, which require multiple queries to models like GPT for comprehensive descriptions.

3. The study on diverse semantics and the importance of multiple learnable prompts is well-motivated.

4. The extensive few-shot learning experiments and comprehensive ablation study effectively demonstrate the proposed methods and each module.

**Weaknesses:**

1. Although the paper demonstrates superior few-shot performance, the notable aspect is the open-vocabulary ability of CLIP. Since 2023, literature has focused on improving this aspect. Additional experiments on this topic are suggested.

2. The authors should discuss whether the proposed distillation method works when ground-truth training data is missing. Solely text-side fine-tuning may lead to misalignment with the visual side. If it doesn't work, this limitation should be addressed.

3. The computational costs, including time and memory during training and inference, should be compared to existing methods. This is especially important because multiple prompts can introduce substantial costs, particularly when dealing with a large number of classes, such as in ImageNet-1K.

4. The symbol $t$ is defined both as prompt embeddings (L198) and learnable prompt vectors (L202); these should be differentiated.

**Questions:**

In line 7 of Algorithm 1, why do the text embeddings $g(c_i)$ from the output of the CLIP text encoder go through the encoder again?

---

> ### Author Response · Authors · 2024-11-24
> **Response for RgBn - Part 1**
>
> Thank you for your thorough analysis and constructive feedback on our paper. Here are our responses to the four weaknesses and one question you pointed out. We hope these answers address your concerns to some extent, and we would be happy to engage in further discussions if you have additional questions or comments.
>
> **Response 1.**
> > Although the paper demonstrates superior few-shot performance, the notable aspect is the open-vocabulary ability of CLIP. Since 2023, literature has focused on improving this aspect. Additional experiments on this topic are suggested.
>
> Following your comment, we conducted Base-to-Novel generalization experiments inspired by the MaPLe[a] paper. Since we were unable to replicate the exact experimental settings used in MaPLe, we devised our own setup as follows: the dataset was split into Base and Novel classes in a 7:3 ratio (random split). We trained the model on the Base classes with a 16-shot setting and evaluated the generalization performance on the Novel classes.
>
> * Exp on DTD dataset
>   |       | Base | Novel |
>   |-------|------------|-------------|
>   | CoOp  | 73.5       | 39.9        |
>   | MaPLe | 76.3       | 42.6        |
>   | **DeMul (Ours)** | **78.1**   | **45.0** |
>
> * Exp on EuroSAT dataset
>   |       | Base | Novel |
>   |-------|---------------|-----------------|
>   | CoOp  | 89.2           | 64.2           |
>   | MaPLe | 92.4           | 67.8           |
>   | **DeMul (Ours)** | **93.8**       | **69.0**       |
>
>   Although we could only evaluate performance on two datasets within the given timeframe, we believe these results suggest that our approach may also be effective for open-vocabulary tasks. We hope this experiment provides clarity and valuable insights to the issue raised.
>
>   [a] MaPLe: Multi-modal Prompt Learning, CVPR 2023
>
> **Response 2.**
> >The authors should discuss whether the proposed distillation method works when ground-truth training data is missing. Solely text-side fine-tuning may lead to misalignment with the visual side. If it doesn't work, this limitation should be addressed.
>
> We have not explored unsupervised scenarios without ground truth in this paper. At present, we have not provided concrete results or justifications for the two directions you pointed out, but we will further reflect on them and include any insights in the limitations section. However, we would like to share some of our thoughts on the issue of misalignment with the visual side below. Thank you for offering valuable suggestions regarding the future directions of our research:
>
> Basically, our work follows the fine-tuning methodology. As such, we did not train VLM models from scratch using image data, and thus we believe we did not encounter the misalignment problem you mentioned. Additionally, during the model design process, we faced constraints in obtaining new semantics by transforming images in tasks like few-shot learning. This led us to focus on freely transforming the semantics of text and exploring how to distill the capabilities of LLMs.
>
> That said, we find your question highly relevant. While this is not a complete answer, we would like to share some observations regarding the features of CLIP's text encoder.
>
> For the pre-trained text encoder \\( g \\) of CLIP, we sampled \\( x_i \sim \mathcal{N}(0, 0.02) \\) for \\( 1 \leq i \leq 1000 \\).
>
>   **Experiment 1:**
>   - Computed sequences \\(\\{g^{(j)}(x_i)\\}_{j=0}^{40}\\) for \\(1 \\leq i \\leq 1000\\).
>   - Observed that each sequence \\( \\{ ||\\{g^{(j)}(x_i)||\_2\\}_{j=0}^{40} \\) is monotonically increasing.
>   - Found that when \\(||x||_2 < 20\\), \\(\cos\angle(x, g(x)) \leq 0.2\\), and when \\(||x||_2 > 100\\), \\(\cos\angle(x, g(x)) \approx 1\\).
>
>   From this experiment, we observed that \\(g\\) preserves the angle of a point in \\(\mathbb{R}^{512}\setminus B_R(0)\\) for some \\(R > 0\\).
>
>   **Experiment 2:**
>   - Selected a dummy class \\(c \in \mathbb{R}^{512}\setminus B_{100}(0)\\).
>   - Concatenated \\(c\\) to samples to create \\(c \oplus x_i \sim \mathcal{N}(c \oplus 0, 0.02)\\) for \\(1 \leq i \leq 1000\\).
>   - Computed \\(\\{g(c \oplus x_i)\\}_{i=1}^{1000}\\) and visualized its 2D PCA projection.
>
> From this experiment, we observed that the variance of \\(\\{g(c \oplus x_i)\\}_{i=1}^{1000}\\)  is larger than. \\( \\{c \oplus x_i\\}\_{\{i=1\}}^{1000} \\) .
>
>   These experiments suggest that beyond a certain radius \\(R > 0\\), the text encoder \\(g\\) slightly deforms the distribution into a cone shape while preserving the angles. Due to this property, we conclude that the prompts learned by DeMul for an initially given representation of a class do not significantly alter the class's alignment but instead perturb it slightly to generate different semantics.

---

> ### Author Response · Authors · 2024-11-24
> **Response for RgBn - Part 2**
>
> **Response 3.**
> > The computational costs, including time and memory during training and inference, should be compared to existing methods. This is especially important because multiple prompts can introduce substantial costs, particularly when dealing with a large number of classes, such as in ImageNet-1K.
>
> We understand that your concern may be related to the potential increase in computational costs as diverse prompts are learned. However, our approach does not differ significantly in terms of computational costs compared to existing multi-prompt methods. As highlighted in the limitations section of our conclusion, there is an inherent trade-off between learning class-specific prompts and maintaining memory efficiency. Our approach prioritizes memory efficiency to effectively address the limitations of existing LLM distillation methods, and that's why we maintained the shared prompts across classes. Exploring improvements in class-specific prompts could serve as a meaningful future research direction, as it represents a distinct and substantial area of study.
>
>  Since DeMul does not utilize class-specific prompts, our method operates in the same multi-prompt setting as ProDA[b], which employs shared prompts in a similar manner. Consequently, the computational costs in terms of time and memory are equivalent to those of multi-prompt methods. While the different loss terms introduced in DeMul may slightly increase time consumption, this increase is minimal and does not significantly impact overall efficiency.
>
> We hope this response addresses your concern and clarifies our approach effectively.
>
> [b] Prompt Distribution Learning, CVPR 2022
>
> **Response 4.**
> > The symbol is defined both as prompt embeddings (L198) and learnable prompt vectors (L202); these should be differentiated.
>
> The advice seems to be to distinguish between the embedding vector used to pretrain the mapping network and the soft prompt in the few-shot task. We think this is relevant advice for us. We had a lot of trouble with this, but unlike \\(\psi\\), which is fixed after pretraining, the mapping network \\(\varphi\\), which is learned, is soft-prompted in the few-shot task, so we unified the symbols. However, since we described \\(t_i\\) as an element of \\(\mathcal{D}_{\text{mapping}}\\), it seems better to modify it as advised. We modified the symbol in equation (2), and the revised texts are highlighted in \\(\textcolor{violet}{\text{violet}}\\) color.
>
> **Response for Questions-1.**
> > In line 7 of Algorithm 1, why do the text embeddings from the output of the CLIP text encoder go through the encoder again?
>
>   The mapping \\(p_{\ast}(\\{v\_i\\}\_{\{i=1\}}^L) = [v\_1][v\_2]\cdots [v\_L]\\) is defined by the concatenation of vectors \\(v_i \in \mathbb{R}^d\\) for \\(1 \leq i \leq L\\). We use the first embedding \\(g(c_i)\\) as a vector representation of class \\(c_i\\) to imbue the class information into the (soft) prompt \\( p\_{\ast}(V^{(\\tau)}\_j, g(c\_i)) \\) in line 7 of Algorithm 1. By the additional experiments above, \\(g(p\_{\ast}(V^{(\\tau)}\_j, g(c\_i)))\\) can be considered as \\(g(c\_i) + \\varepsilon\\) for some perturbation \\(\\varepsilon\\).

---

### Author Response · Authors · 2024-11-24
**General comment**

We sincerely thank our reviewers for their detailed feedback, thoughtful reviews, and insightful comments. Based on the received reviews, we have incorporated all feasible changes and uploaded a revised version of the paper. (The revised texts are highlighted in **violet** color.) If you feel further modifications are necessary, we would greatly appreciate any additional comments.

The updated content is as follows:

- Updated the notation in equation (2) on line 197 by replacing \\(t\\) with \\(e\\) to avoid reuse of \\(t\\) in different contexts.
- Added a detailed explanation of the algorithm to provide better clarity.
- Emphasized in the caption of Figure 5 (top image) that it represents relative changes from the initialized values, rather than absolute values.
- Expanded the Future Works section to address limitations highlighted in the reviews and outline potential research directions.
- Added missing citations.

Moreover, some questions were raised by multiple reviewers, and we provide additional experimental results to address these points below.

---

### 1. Ablation Study on Fixed Values \\(\alpha\\) and \\(\lambda\\)
(For reviewers **EZLZ, JO4M, ONAD**)

One concern was the absence of an ablation study for the fixed values of \\(\alpha\\) and \\(\lambda\\) used in our experiments. Below, we present the results of the ablation study conducted on the DTD dataset in the 1-shot setting, which helped us determine the optimal values.

1. **\\(\alpha\\):**
   We experimented with multiple values of \\(\alpha\\) to balance the classification loss and the distillation loss. Notably, setting \\(\alpha = 0.01\\) or \\(\alpha = 0.1\\) led to a significant performance drop. This suggests that prompt weight regularization was either too weak (\\(\alpha = 0.01\\)) or too dominant (\\(\alpha = 0.1\\)), disrupting the balance between the two losses. The best performance was achieved with \\(\alpha = 0.05\\).

2. **\\(\lambda\\):**
   We explored \\(\lambda\\) values from 0.1 to 1.0 to evaluate the effect of L1 regularization on prompt weighting. The results show that \\(\lambda = 0.4 \sim 0.5\\) yielded the best performance. A smaller \\(\lambda\\) (e.g., 0.1) reduces the impact of distillation, while a larger \\(\lambda\\) (e.g., 1.0) overly emphasizes the regularization, diminishing the classification accuracy. This demonstrates that a balanced degree of distillation and classification is critical for optimal performance.

**Ablation Study Results**
|                | 0.1  | 0.2  | 0.3  | 0.4  | **0.5** | 0.6  | 0.7  | 0.8  | 0.9  | 1.0  |
|----------------|-------|-------|-------|-------|---------|-------|-------|-------|-------|-------|
| **\\(\alpha = 0.01\\)** | 49.5  | 50.3  | 49.0  | 57.7  | 53.3    | 50.0  | 52.1  | 49.1  | 49.9  | 53.4  |
| **\\(\alpha = 0.05\\)** | 47.8  | 51.2  | 59.3  | 61.5  | **62.1** | 60.2  | 61.3  | 58.2  | 59.9  | 60.0  |
| **\\(\alpha = 0.1\\)**  | 42.5  | 45.6  | 49.0  | 47.0  | 44.7    | 50.1  | 47.3  | 49.1  | 51.3  | 48.4  |

We believe these results adequately justify the fixed values used in the paper (\\(\lambda = 0.5\\), \\(\alpha = 0.05\\)).

---

### 2. Dependency on Training Data Distribution
(For reviewers **EZLZ, 3ZUA**)

Another concern was the extent to which our method depends on the distribution of training data. As mentioned in the Conclusion, one limitation of our method is its dependency on the training image distribution, which remains an inherent challenge in few-shot learning. To ensure the reliability of the reported scores, we averaged the results from multiple experiments. For all experimental settings, the training data was randomly sampled, and we used the average performance from approximately 5-10 runs.

**Results of 10 Experiments on DTD Dataset**
| \#shot         | 1     | 2     | 3     | 4     | 5     | 6     | 7     | 8     | 9     | 10    | **Avg**  |
|-----------------|-------|-------|-------|-------|-------|-------|-------|-------|-------|-------|----------|
| **1-shot**     | 62.3  | 61.9  | 62.6  | 61.9  | 62.5  | 61.0  | 62.3  | 61.4  | 62.5  | 62.3  | **62.1** |
| **2-shot**     | 66.9  | 67.3  | 68.6  | 67.8  | 67.0  | 68.9  | 67.9  | 67.5  | 68.3  | 68.5  | **67.9** |
| **4-shot**     | 70.2  | 69.8  | 71.0  | 71.2  | 69.2  | 70.5  | 70.4  | 69.9  | 70.1  | 71.1  | **70.3** |
| **8-shot**     | 74.9  | 75.2  | 75.1  | 75.2  | 75.6  | 74.9  | 75.2  | 75.3  | 74.8  | 74.5  | **75.1** |
| **16-shot**     | 76.5  | 77.0  | 77.0  | 78.0  | 76.2  | 76.9  | 78.2  | 76.8  | 76.6  | 76.9  | **77.0** |

---

> ### Author Response · Authors · 2024-11-26
> **Uploaded Final PDF and a Gentle Reminder**
>
> Dear Reviewers,
>
> Previously, we highlighted the revised sections in violet to help reviewers easily identify the changes. However, since the deadline for modifying the paper’s PDF is tomorrow (by November 26), we have decided to revert all text colors back to black for consistency and readability across the entire paper.
>
> Other than **changing the text color from violet back to black, no further revisions have been made**. To ensure transparency, we have documented the previously highlighted revisions below for your reference, in case any reviewer has not yet had the chance to review them.
>
> The sections previously highlighted in violet are as follows:
> - Updated the notation in equation (2) on line 197 by replacing \\(t\\) with \\(e\\) to avoid reuse of \\(t\\) in different contexts.
>   - Equation (2) on line 197.
> - Added a detailed explanation of the algorithm to provide better clarity.
>   - Lines 321–370.
> - Emphasized in the caption of Figure 5 (top image) that it represents relative changes from the initialized values, rather than absolute values.
>   - Figure 5 caption on line 463
>   - Original: “(Top) illustrates the variations in weights corresponding to each prompt.”
>   - Revised: “(Top) illustrates the variations in weights corresponding to each prompt by their relative changes from their initialized values.”
> - Expanded the Future Works section to address limitations highlighted in the reviews and outline potential research directions.
>   - Two new sentences starting from “Moreover~” on line 537.
> - Added missing citations.
>   - Included additional citations in the Preliminaries and Background section.
>
>
> Also, we understand that November 26 is the final date by which reviewers can provide feedback. **If you have any further comments or suggestions**, we would greatly appreciate. We will do our best to address them promptly.
>
> **Once again, we sincerely thank all reviewers for their thoughtful and constructive feedback.**
>
> Sincerely,
> The Authors

---

### Meta-Review · Area_Chair_YQP2 · 2024-12-18

**Metareview:**

The paper addresses prompt learning in VLMs by directly distilling the knowledge in a GPT model to the learnable prompt vectors in VLMs. The method essentially maps the learnable prompt vectors to the GPT's embedding space and achieves significant improvements over previous prompt learning methods. The paper received five reviews with 4x borderline accept and 1x accept. The reviewers generally found the idea interesting and novel, and the results strong. In terms of concerns, they raised questions about the computational cost and some design choices, and also requested more results under different experimental settings. The rebuttal well addressed these questions. After reading the paper, the AC agrees with the reviewers that the paper has merits and therefore recommends acceptance.

**Additional Comments On Reviewer Discussion:**

The questions raised during the review are mainly about confusions on some concepts and requests for more results under different problem settings. The authors have properly addressed these issues in the rebuttal.

---

### Decision · Program_Chairs · 2025-01-22

Accept (Poster)